# GENERALIZED UNIVERSAL APPROXIMATION FOR CERTIFIED NETWORKS

## ABSTRACT

To certify safety and robustness of neural networks, researchers have successfully applied *abstract interpretation*, primarily using *interval bound propagation*. To understand the power of interval bounds, we present the *abstract universal approximation* (AUA) theorem, a generalization of the recent result by Baader et al. (2020) for ReLU networks to a large class of neural networks. The AUA theorem states that for any continuous function $f$, there exists a neural network that (1) approximates $f$ (universal approximation) and (2) whose interval bounds are an arbitrarily close approximation of the set semantics of $f$. The network may be constructed using *any* activation function from a rich class of functions—sigmoid, tanh, ReLU, ELU, etc.—making our result quite general. The key implication of the AUA theorem is that there always exists certifiably robust neural networks, which can be constructed using a wide range of activation functions.

## 1 INTRODUCTION

With wide adoption of neural networks, new safety and security concerns arose. The most prominent property of study has been *robustness* (Goodfellow et al., 2015): small perturbations to the input of a network should not change the prediction. For example, a small change to an image of a stop sign should not cause a classifier to think it is a speed-limit sign. A number of researchers have proposed the use of *abstract interpretation* (Cousot & Cousot, 1977) techniques to prove robustness of neural networks (Gehr et al., 2018; Wang et al., 2018; Anderson et al., 2019) and to train robust models (Mirman et al., 2018; Gowal et al., 2018; Huang et al., 2019; Wong & Kolter, 2018; Wong et al., 2018; Balunovic & Vechev, 2020).

Suppose we want to verify robustness of a neural network to small changes in the brightness of an image. We can represent a large set of images, with varying brightness, as an element of some *abstract domain*, and propagate it through the network, effectively executing the network on an intractably large number of images. If all images lead to the same prediction, then we have a proof that the network is robust on the original image. The simplest abstract interpretation technique that leads to practical verification results is *interval analysis*—also referred to as *interval bound propagation*. In our example, if each pixel in a monochrome image is a real number $r$, then the pixel can be represented as an interval $[r - \epsilon, r + \epsilon]$, where $\epsilon$ denotes the range of brightness we wish to be robust on. Then, the *box* representing the interval of each pixel is propagated through the network using interval arithmetic operations.

The interval domain has been successfully used for certifying properties of neural networks in vision (Gehr et al., 2018; Gowal et al., 2018), NLP (Huang et al., 2019), as well as cyber-physical systems (Wang et al., 2018). Why does the interval domain work for certifying neural networks?

To begin understanding this question, Baader et al. (2020) demonstrated a surprising connection between the *universal approximation theorem* and neural-network certification using interval bounds. Their theorem states that not only can neural networks approximate any continuous function $f$ (universal approximation) as we have known for decades, *but* we can find a neural network, using *rectified linear unit* (ReLU) activation functions, whose interval bounds are an arbitrarily close approximation of the *set* semantics of $f$, i.e., the result of applying $f$ to a set of inputs (e.g., set of similar images).

**AUA theorem (semi-formally):** For a continuous function $f : \mathbb{R}^m \to \mathbb{R}$ that we wish to approximate and error $\delta > 0$, there is a neural network $N$ that has the following behavior:

Let $B \subset \mathbb{R}^m$ be a box. The red interval (top) is the tightest interval that contains all outputs of $f$ when applied to $\mathbf{x} \in B$.

If we propagate box $B$ through $N$ using interval bounds, we may get the black interval (bottom) $N^\#(B)$, whose lower/upper bounds are up to $\delta$ away from the red interval.

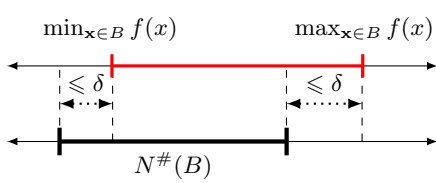

Figure 1: Semi-formal illustration of AUA theorem. (Right is adapted from Baader et al. (2020).)

The theorem of Baader et al. (2020) is restricted to networks that use rectified linear units (ReLU). In this work, we present a general universal approximation result for certified networks using a rich class of well-behaved activation functions. Specifically, we make the following contributions.

**Abstract universal approximation (AUA) theorem.** We prove what we call the *abstract universal approximation theorem*, or AUA theorem for short: Let $f$ be the function we wish to approximate, and let $\delta > 0$ be the tolerated error. Then, there exists a neural network $N$, built using *any* well-behaved activation function, that has the following behavior: For any box of inputs $B$, we can certify, using interval bounds, that the range of outputs of $N$ is $\delta$ close to the range outputs of $f$. If the box $B$ of inputs is a single point in Euclidean space, the AUA theorem reduces to the universal approximation theorem; thus, AUA generalizes universal approximation. *Fig. 1 further illustrates the AUA theorem.*

**Existence of robust classifiers.** While the AUA theorem is purely theoretical, it sheds light on the existence of certifiable neural networks. Suppose there is some ideal robust image classifier $f$ using the $\ell_\infty$ norm, which is typically used to define a set of images in the neighborhood of a given image. The classical universal approximation theorem tells us that, for any desired precision, there is a neural network that can approximate $f$. We prove that the AUA theorem implies us that there exists a neural network for which we can automatically certify robustness using interval bounds while controlling approximation error. In addition, this neural network can be built using almost any activation function in the literature, and more.

**Squashable functions.** We define a rich class of activation functions, which we call *squashable* functions, for which our abstract universal approximation theorem holds. This class expands the functions defined by Hornik et al. (1989) for universal approximation and includes popular activation functions, like ReLU, sigmoid, tanh, ELU, and other activations that have been shown to be useful for training robust neural networks (Xie et al., 2020). The key feature of squashable activation functions is that they have left and right limits (or we can use them to construct functions with limits). We exploit limits to approximate step (sign) functions, and therefore construct step-like approximations of $f$, while controlling approximation error $\delta$.

**Proof of AUA theorem.** We present a constructive proof of the AUA theorem. Our construction is inspired by and synthesizes a range of results: (1) the work of Hornik et al. (1989) on *squashing* functions for universal approximation, (2) the work of Csáji (2001) for using the sign (step) function to construct *Haar* (wavelet) functions, and (3) the work of Baader et al. (2020) on the specialized AUA theorem for ReLUs. The key idea of Baader et al. (2020) is to construct an indicator function for box-shaped regions. We observe that squashable functions can approximate the sign function, and therefore approximate such indicator functions, while carefully controlling precision of abstract interpretation. Our proof uses a simpler indicator construction compared to Baader et al. (2020), and as a result its analysis is also simpler.

## 2 RELATED WORK

The classical universal approximation (UA) theorem has been established for decades. In contrast to AUA, UA states that a neural network with one single hidden layer can approximate any continuous function on a compact domain. One of the first versions goes back to Cybenko (1989); Hornik et al. (1989), who showed that the standard feed-forward neural network with sigmoidal or squashing activations is a universal approximator. The most general version of UA was discovered by Leshno et al. (1993), who showed that the feed-forward neural network is a universal approximator if and only if the activation function is non-polynomial. Because AUA implies UA, this means AUA cannot

**Squashable activation functions that satisfy Eq. (1)**

$$\sigma(x) = \frac{1}{1 + e^{-x}} \qquad \tanh(x) = \frac{2}{1 + e^{-2x}} - 1 \qquad \mathrm{softsign}(x) = \frac{x}{1 + |x|}$$

**Squashable activation functions that do not directly satisfy Eq. (1)**

$$\mathrm{ReLU}(x) = \left\{ \begin{array}{ll} x, & x \geqslant 0 \\ 0, & x < 0 \end{array} \right. \qquad \mathrm{ELU}(x) = \left\{ \begin{array}{ll} x, & x \geqslant 0 \\ e^x - 1, & x < 0 \end{array} \right.$$

$$\mathrm{softplus}(x) = \log(1 + e^x) \qquad \mathrm{smoothReLU}_a(x) = \left\{ \begin{array}{ll} x - \frac{1}{a} \log(ax + 1), & x \geqslant 0 \\ 0, & x < 0 \end{array} \right.$$

Figure 2: Examples of squashable activation functions, including popular functions, and more recent ones: *sigmoid*, *tanh*, *rectified linear units* (ReLU) (Nair & Hinton, 2010), *exponential linear unit* (ELU) (Clevert et al., 2016), *softplus* (Glorot et al., 2011), *softsign* (Bergstra et al., 2009), and *smooth ReLU* (Xie et al., 2020), which is parameterized by $a > 0$.

hold beyond non-polynomial activation functions. There are also other variants of UA. Some of them study the expressiveness of neural networks with structural constraints, such as restricted width per layer Lu et al. (2017); Kidger & Lyons (2019), or specific neural network architectures Lin & Jegelka (2018). Another line of work focuses on specific functions that one wants to approximate rather than arbitrary continuous functions, such as Anil et al. (2019); Cohen et al. (2019), who study approximation of Lipschitz functions.

Neural-network verification has received a lot of attention in recent years. Most techniques are either based on decision procedures, like SMT solvers Ehlers (2017); Katz et al. (2017) and integer linear programming (ILP) solvers Tjeng et al. (2019), or abstract interpretation. The former class can often provide sound and complete verification on neural networks with piecewise-linear operations, like ReLU, but is not scalable due to the complexity of the problem and the size of the networks. Abstract-interpretation-based techniques sacrifice completeness for efficient verification. We have considered the simplest non-trivial numerical domain, intervals, that has been shown to produce strong results, both for robustness verification and adversarial training Gehr et al. (2018); Anderson et al. (2019); Huang et al. (2019); Mirman et al. (2018); Wang et al. (2018); Zhang et al. (2020). Researchers have considered richer domains Singh et al. (2018; 2019), like zonotopes Ghorbal et al. (2009) and forms of polyhedra Cousot & Halbwachs (1978). Since such domains are strictly more precise than intervals, the AUA theorem holds for them.

# 3 FOUNDATIONS AND SQUASHABLE ACTIVATION FUNCTIONS

## 3.1 NEURAL NETWORKS AND SQAUSHABLE ACTIVATIONS

A neural network $N$ is a function in $\mathbb{R}^m \to \mathbb{R}$. We define a network $N$ following a simple grammar, a composition of primitive arithmetic operations and *activation functions*.

**Definition 3.1** (Neural network grammar). *Let $\mathbf{x} \in \mathbb{R}^m$ be the input to the neural network. A neural network $N$ is defined using the following grammar: $N = c \mid x_i \mid N_1 + N_2 \mid c * N_1 \mid t(N_1)$, where $c \in \mathbb{R}$, $x_i$ is the ith element of $\mathbf{x}$, and $t : \mathbb{R} \to \mathbb{R}$ is an activation function. We will always fix a single activation function $t$ to be used in the grammar.*

We now present a general class of activation functions that we will call *squashable* activation functions. Fig. 2 shows some examples.

**Definition 3.2** (Squashable functions). *$t : \mathbb{R} \to \mathbb{R}$ is squashable iff (1) there is $a < b \in \mathbb{R}$ such that*

$$\lim_{x \to -\infty} t(x) = a, \qquad \lim_{x \to \infty} t(x) = b, \quad \text{and} \quad \forall x < y. \, t(x) \leqslant t(y) \tag{1}$$

*or (2) we can construct a function $t'$ that satisfies Eq. (1) as affine transformations and function compositions of copies of $t$, i.e., following the grammar in Def. 3.1. E.g., $t'(x) = t(1 - t(x))$.*

Informally, an activation function is in this class if we can use it to construct a monotonically increasing function that has limits in the left and right directions, $-\infty$ and $\infty$.[1] Squashable activation

---

[1]In our construction and proof, we do not need the function to be monotonic; however, in practice, most activation functions are monotonic and abstractly interpreting arbitrary functions is impractical.

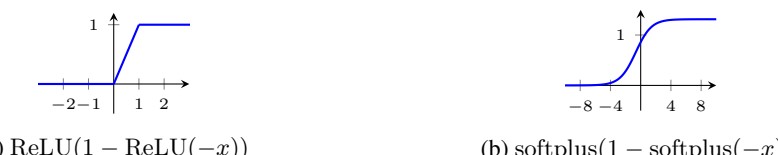

(a) $\mathrm{ReLU}(1 - \mathrm{ReLU}(-x))$        (b) $\mathrm{softplus}(1 - \mathrm{softplus}(-x))$

Figure 3: Two activation functions after applying construction in Proposition 3.3. Observe that the resulting function satisfies Eq. (1), and therefore ReLU and softplus are squashable.

functions extend the *squashing* functions used by Hornik et al. (1989). All of the activation functions in Fig. 2 are squashable.

Fig. 2 (top) shows activation functions that satisfy Eq. (1), and are therefore squashable. For example, sigmoid and tanh easily satisfy Eq. (1): both have limits and are monotonically increasing.

What about activation functions like ReLU, ELU, softplus, etc., shown in Fig. 2 (bottom)? It is easy to see that they do not satisfy Eq. (1): none of them have a right limit. However, by point (2) of Def. 3.2, given an activation function $t$, if we can construct a new activation function $t'$ that satisfies Eq. (1), using the operations in the grammar in Def. 3.1, then $t$ is squashable. We give a general and simple construction that works for all activation functions in Fig. 2 (bottom).

**Proposition 3.3.** *Let* $t \in \{\mathrm{ReLU}, \mathrm{softplus}, \mathrm{smoothReLU}_a, \mathrm{ELU}\}$. *The function* $t'(x) = t(1 - t(-x))$ *satisfies Eq.* (1). *Therefore, ReLU, softplus, Smooth ReLU, and ELU, are squashable.*

**Example 3.4.** *Fig. 3 shows* $t(1 - t(-x))$, *for* $t = \mathrm{ReLU}$ *and* $t = \mathrm{softplus}$. *Both have left/right limits and are monotonic. Thus, they satisfy Eq.* (1) *and therefore ReLU and softplus are squashable.*

## 3.2 Interval Analysis of Neural Networks

Given $f : \mathbb{R}^m \to \mathbb{R}$ and set $S \subseteq \mathbb{R}^m$, we will use $f(S)$ to denote $\{f(\mathbf{x}) \mid \mathbf{x} \in S\}$.

We now define interval versions of the operations of a neural network, which are known as *abstract transformers*. This was first introduced in Cousot & Cousot (1977), which also proved the soundness of the interval domain. An $m$-dimensional box $B$ is a tuple of intervals $[l_1, u_1] \times \ldots \times [l_m, u_m]$. All our operations are over scalars, so we define abstract transformers over 1D boxes.

**Definition 3.5** (Arithmetic abstract transformers). *Let $B$ be an $m$-dimensional box input to the neural network. We follow the grammar in Def. 3.1 to define the abstract transformers.*

$$
\begin{aligned}
c^\# &= [c, c] \\
x_i^\# &= [l_i, u_i], \qquad \textit{where } l_i, u_i \textit{ are the ith lower and upper bounds of } B \\
[l_1, u_1] +^\# [l_2, u_2] &= [l_1 + l_2, u_1 + u_2] \\
[c, c] *^\# [l, u] &= [\min(c * l, c * u), \max(c * l, c * u)]
\end{aligned}
$$

**Definition 3.6** (Abstract transformer for activations). *Let $B = [l, u]$. Then, $t^\#(B) = [t(l), t(u)]$.*

This transforer was introduced in (Gehr et al., 2018).

# 4 Abstract Universal Approximation Theorem & Its Implications

In this section, we state the *abstract universal approximation* (AUA) theorem and its implications.

Assume a fixed continuous function $f : C \to \mathbb{R}$, with a compact domain $C \subset \mathbb{R}^m$, that we wish to approximate.

**Definition 4.1** ($\delta$-abstract approximation). *Let $\delta > 0$. A neural network $N$ $\delta$-abstractly approximates $f$ iff for every box $B \subseteq C$, we have $[l + \delta, u - \delta] \subseteq N^\#(B) \subseteq [l - \delta, u + \delta]$, where $l = \min f(B)$ and $u = \max f(B)$.*

$\delta$-abstract approximation says that the box output of abstract interpretation $N^\#(B)$ is up to $\delta$ away from the tightest bounding box around the set semantics $f(B)$. It was developed in Baader et al. (2020). Observe that the standard notions of approximation is a special case of $\delta$-abstract approximation, when the box $B$ is a point in $C$.

We now state our main theorem:

> **Theorem 4.2** (Abstract universal approximation). *Let $f : C \to \mathbb{R}$ be a continuous function on compact domain $C \subset \mathbb{R}^m$. Let $t$ be a squashable activation function. Let $\delta > 0$. There exists a neural network $N$, using only activations $t$, that $\delta$-abstractly approximates $f$.*

Informally, the theorem says that we can always find a neural network whose abstract interpretation is arbitrarily close to the set semantics of the approximated function. Note also that there exists such a neural network for any fixed squashable activation function $t$. In the appendix, we give a generalization of the AUA theorem to functions and networks with multiple outputs.

As we discuss next, the AUA theorem has very exciting implications: We can show that one can always construct provably *robust* neural networks using any squashable activation function (Thm. 4.5).

We begin by defining a robust classifier in $\ell_\infty$ norm. We treat $f : C \to \mathbb{R}$ as a *binary* classifier, where an output $< 0.5$ represents one class and $\geqslant 0.5$ represents another.

**Definition 4.3** ($\epsilon$-Robustness). *Let $\mathbf{x} \in \mathbb{R}^m$, $M \subseteq C$ and $\epsilon > 0$. The $\epsilon$-ball of $\mathbf{x}$ is $R_\epsilon(\mathbf{x}) = \{\mathbf{z} \mid \|\mathbf{z} - \mathbf{x}\|_\infty \leqslant \epsilon\}$. We say that $f$ is $\epsilon$-robust on set $M$ iff, for all $\mathbf{x} \in M$ and $\mathbf{z} \in R_\epsilon(\mathbf{x})$, we have $f(\mathbf{x}) < 0.5$ iff $f(\mathbf{z}) < 0.5$.*

**Definition 4.4** (Certifiably robust networks). *A neural network $N$ is $\epsilon$-certifiably robust on $M$ iff, for all $\mathbf{x} \in M$, we have $N^\#(B) \subseteq (-\infty, 0.5)$ or $N^\#(B) \subseteq [0.5, \infty)$, where $B = R_\epsilon(\mathbf{x})$ (Note that an $\epsilon$-ball is a box in $\mathbb{R}^m$.).*

From an automation perspective, the set $M$ is typically a finite set of points, e.g., images. For every $\mathbf{x} \in M$, the verifier abstractly interprets $N$ on the $\epsilon$-ball of $\mathbf{x}$, deriving a lower bound and upper bound of the set of predictions $N(R_\epsilon(\mathbf{x}))$. If the lower bound is $\geqslant 0.5$ or the upper bound is $< 0.5$, then we have proven that all images in the $\epsilon$-ball have the same classification using $N$.

Assuming there is some ideal robust classifier, then, following the AUA theorem, we can construct a neural network, using any squashable activation function, that matches the classifier's predictions and is certifiably robust. Refer to the supplementary materials for an extension to $n$-ary classifiers.

**Theorem 4.5** (Existence of robust networks). *Let $f : C \to \mathbb{R}$ be $\epsilon$-robust on set $M \subseteq C$. Assume that $\forall \mathbf{x} \in M, \mathbf{z} \in R_\epsilon(\mathbf{x}). f(\mathbf{z}) \neq 0.5$.[2] Let $t$ be a squashable activation function. Then, there exists a neural network $N$, using activation functions $t$, that (1) agrees with $f$ on $M$, i.e., $\forall \mathbf{x} \in M. N(\mathbf{x}) < 0.5$ iff $f(\mathbf{x}) < 0.5$, and (2) is $\epsilon$-certifiably robust on $M$.*

## 5 PROOF OF AUA THEOREM: AN OVERVIEW

We now give an overview of our proof of the AUA theorem, focusing on its novelty. Our proof is constructive: we show how to construct a neural network that $\delta$-abstractly approximates a function $f$.

**Approximating indicator functions (Sec. 6).** The key element, and primary novelty of our proof, is the construction of indicator functions from squashable functions. Our construction is motivated by Csáji (2001), who used sign functions to construct the Haar function, which can be used to uniformly approximate any continuous function. See Fig. 5b for an example of the indicator function's approximation with a sigmoid.

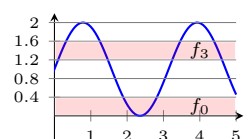

(a) Sliced $sin(2x) + 1$ with range of size $0.4$

It is a classical idea to use indicator functions to approximate a continuous function in a piecewise fashion—see Nielsen (2015, Ch.4) for an interactive visualization of universal approximation. However, for AUA, the input to the function is a box, and therefore we need to make sure that our indicator-function approximations provide tight *abstract* approximations (Def. 4.1), as will be shown in Thm. 6.2.

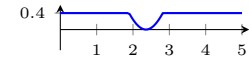

(b) Example slice $f_0$

Figure 4: Slicing example

---

[2]This assumption eliminates the corner case where a point sits exactly on the classification boundary, $0.5$.

**Composing indicator functions (Sec. 7).** Once we have approximated indicator functions, we need to put them together to approximate the entire function $f$. The remainder of the construction is an adaption of one by Baader et al. (2020) for ReLU networks.

The construction starts by slicing $f$ into a sequence of functions $f_i$ such that $f = \sum_i f_i$. Each $f_i$ captures the behavior of $f$ on a small interval of its range. See Fig. 4 for an example of slicing. Next, we approximate each $f_i$ using a neural network $N_i$. Slicing ensures that $\delta$-abstract approximation is tight for large boxes. Because $f(B) = \sum_i f_i(B)$, when approximating $f_i(B)$ using $N_i(B)$, we can show that for most $i$, $f_i(B) \approx N_i(B)$. The smaller the range of slice $f_i$, the tighter the abstract approximation of $f$ using indicator functions.

Our construction and analysis are different from Baader et al. (2020) in the following ways:

1. Baader et al. (2020) focus exclusively on ReLU activations. In our work, we consider squashable functions, which contain most commonly used activation functions. Compared to ReLU, which has rigid values, we only know that squashable functions have limits at both sides. This makes the class of functions more expressive but also harder to quantify. When analyzing the interval bounds of the whole network construction, we need to take into account the extra imprecision, which propagates from the indicator function to the whole neural network.

2. The key construction of Baader et al. (2020) is to build $\min(x_1, x_2, \ldots, x_{2m})$ using ReLUs. The depth of the construction depends on $m$, and the analysis of its interval bounds is rather complicated. Our construction uses only two layers of activations, resulting in a much simpler analysis. Because ReLU is a squashable function, a by-product is that if we only consider AUA for ReLU, our construction and its analysis are simpler than that of Baader et al. (2020).

## 6 ABSTRACTLY APPROXIMATING INDICATOR FUNCTIONS

We begin by showing the crux of our construction: how to approximate an indicator function. Recall that our goal is to $\delta$-abstractly approximate a continuous function $f : C \to \mathbb{R}$.

**Grid of boxes.** Fix $\epsilon > 0$. Consider a standard *grid* of vertices over $C$, where any two neighboring vertices are axis-aligned and of distance $\epsilon$; we will call this an $\epsilon$-grid.[3] Let $[a_1, b_1] \times \ldots \times [a_m, b_m]$ be a box $G$ on the grid, where $[a_i, b_i]$ is the range of $G$ at dimension $i$. In other words, $b_i - a_i$ is a multiple of $\epsilon$. The neighborhood $\nu(G)$ of $G$ is $[a_1 - \epsilon, b_1 + \epsilon] \times \ldots \times [a_m - \epsilon, b_m + \epsilon]$. Our goal is to construct an indicator function whose value is close to 1 within $G$, and close to 0 outside $G$'s neighborhood $\nu(G)$. The idea of using grid is similar to the nodal basis in He et al. (2018).

**Indicator-function approximation intuition.** Given any activation function $t$ that satisfies Eq. (1), a key observation is that if we dilate $t$ properly, i.e., multiply the input with a large number $\mu$ to get $t(\mu x)$, we will obtain an approximation of the sign (or step) indicator function—a function that indicates whether an input is positive or negative: $\text{sign}(x) = 1$ if $x \geqslant 0$ and 0 otherwise. A sign function can be used to construct an indicator function for 1-dimensional boxes. For example, $\text{sign}(x) - \text{sign}(x - 1)$ returns 1 for $x \in (0, 1]$, and 0 otherwise. In what follows, we will use the above ideas to approximate the sign function and the indicator function for $m$-dimensional boxes.

### 6.1 APPROXIMATING A ONE-DIMENSIONAL INDICATOR FUNCTION

We will first show how to construct an indicator function for a 1D box, using a squashable function. The main challenge is choosing the dilation factor that results in tight abstract approximation.

Without loss of generality, assume we are given a squashable function $t$ that (1) satisfies Eq. (1) and (2) has left and right limits of 0 and 1, respectively.[4]

**Loss of precision from limits.** The activation function $t$ has limits at both sides, but the function might never reach the limit. For example, the right limit of the sigmoid function, $\sigma$, is 1, but $\forall x.\, \sigma(x) \neq 1$. This will lead to a loss of precision when we use $t$ to model a sign function. However, we can carefully apply mathematical analysis to rigorously bound this imprecision.

---

[3]The vertices of the grid form a natural $\epsilon/2$-net on $C$ equipped with $l_\infty$ metric.

[4]If (1) is not satisfied, by Def. 3.2, we can construct a $t'$ that satisfies Eq. (1) from $t$. If (2) is not satisfied, we can apply an affine transformation to the results of $t$ to make the left and right limits 0 and 1.

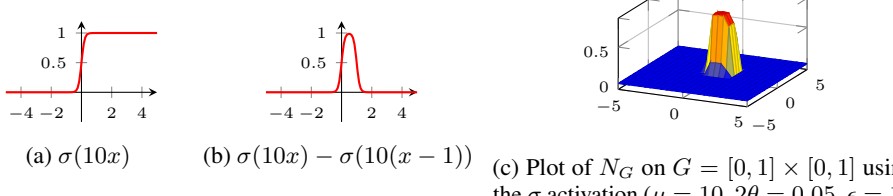

(a) $\sigma(10x)$     (b) $\sigma(10x) - \sigma(10(x-1))$     (c) Plot of $N_G$ on $G = [0,1] \times [0,1]$ using the $\sigma$ activation ($\mu = 10$, $2\theta = 0.05$, $\epsilon = 1$).

Figure 5: Approximating indicator functions on $[0, \infty)$, $[0,1]$ and $[0,1] \times [0,1]$ using the sigmoid activation functions

**Dilation to approximate sign function.** We now discuss how to dilate $t$ to get a sign-function-like behavior. By definition of limit, we know the following lemma, which states that $\forall \theta > 0$ by sufficiently increasing the input of $t$, we can get $\theta$ close to the right limit of 1, and analogously for the left limit. Figs. 5a and 5b show how sigmoid can approximate an indicator function.

Because the grid size is $\epsilon$, we want the sign-function approximation to achieve a transition from $\approx 0$ to $\approx 1$ within $\epsilon$. Let $\mu$ be the *dilation factor*. We would like the following (Fig. 6 illustrates the loss of precision $\theta$ incurred by our construction):

**Lemma 6.1.** $\forall \theta > 0, \exists \mu > 0$ *such that: (1) if $x \geqslant 0.5\epsilon$, then $t(\mu x) \in (1 - \theta, 1]$; (2) if $x \leqslant -0.5\epsilon$, then $t(\mu x) \in [0, \theta)$.*

**Indicator function on dimension $i$.** Recall that the projection of a box $G$ on dimension $i$ is $[a_i, b_i]$. We want to build an indicator function that has value close to 1 on $[a_i, b_i]$, and value close to 0 on $\mathbb{R} \setminus [a_i - \epsilon, b_i + \epsilon]$. Notice that our approximation may not be able to exactly tell if we are in $G$ or its neighborhood.

Inspired by how to construct an indicator function from a sign function, we will take the difference between two shifted sign functions. Let

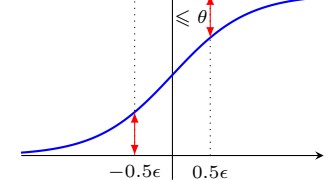

Figure 6: Loss of precision $\theta$ due to use of squashable activation to approximate a sign function. Length of red arrows is $\leqslant \theta$.

$$\hat{t}_i(x) = t\left(\mu\left(x - (a_i - 0.5\epsilon)\right)\right) - t\left(\mu\left(x - (b_i + 0.5\epsilon)\right)\right) \quad (2)$$

We choose the two points $a_i - 0.5\epsilon$ and $b_i + 0.5\epsilon$, which lie in the middle of the $[a_i, b_i]$ and its neighborhood, so that $\hat{t}_i$'s value of is close to 1 within $[a_i, b_i]$, and 0 outside $[a_i, b_i]$'s neighborhood.

## 6.2 APPROXIMATING AN $m$-DIMENSIONAL INDICATOR

We saw how to construct an indicator approximation for a 1-dimensional box. We will now show how to construct an indicator function approximation $N_G$ for an $m$-dimensional box.

**Constructing $N_G$.** We want to construct an indicator function whose value within a box $G$ is close to 1, and outside the neighborhood $\nu(G)$ is close to 0. In the multi-dimensional case, $m \geqslant 2$, we do not know at which, if any, dimension $j$ of an input is outside the neighborhood of $G$. The 1-dimensional indicator approximation, $\hat{t}$, which we constructed earlier, can be used to tell us, for each dimension $j$, whether $x_j$ is within the bounds of the neighborhood of $G$. Therefore we can construct a logical OR approximation that applies $\hat{t}$ to each dimension and takes the OR of the results. Specifically: (1) we will construct a function that applies $\hat{t}$ to each dimension, and sums the results such that the answer is $> 0$ if $\mathbf{x} \in G$, and $< 0$ if $\mathbf{x} \notin \nu(G)$. (2) Then, we can use the sign-function approximation to indicate the sign of the answer.

Formally, we define the neural network $N_G$ as follows:

$$N_G(\mathbf{x}) = t\left(\mu\left(\sum_{i=1}^{m} H_i(x_i) + 0.5\epsilon\right)\right) \quad (3)$$

where $H_i(x) = \hat{t}_i(x) - (1 - 2\theta)$. Eq. (3) has a similar structure to the $m$-dimensional indicator in Baader et al. (2020), i.e., both of them use the activation function to evaluate the information from all dimensions.

The term $\sum_{i=1}^{m} H_i(x_i)$ evaluates to a positive value if $\mathbf{x} \in G$ and to a negative value if $\mathbf{x} \notin \nu(G)$. Observe that we need to shift the result of $\hat{t}$ by $(1 - 2\theta)$ to ensure a negative answer if one of the dimensions is outside the neighborhood. Then, we use $t$ to approximate the sign function, as we did in the 1-dimensional case, giving $\approx 1$ if $\mathbf{x} \in G$, and $\approx 0$ if $\mathbf{x} \notin \nu(G)$. Fig. 5c shows a plot of a two dimensional $N_G$.

**Abstract precision of indicator approximation.** The following key theorem states the precision of the abstract interpretation of $N_G$: if the input box is in $G$, then the output box is within $\theta$ from 1; if $B$ is outside the neighborhood of $G$, then the output box is within $\theta$ from 0.

**Theorem 6.2** (Abstract interpretation of $N_G$). *For any box $B \subseteq C$, the following is true:*

1. $N_G^{\#}(B) \subseteq [0, 1]$.
2. *If $B \subseteq G$, then $N_G^{\#}(B) \subseteq (1 - \theta, 1]$.*
3. *If $B \subseteq C \setminus \nu(G)$, then $N_G^{\#}(B) \subseteq [0, \theta)$.*

**Complexity of construction.** To construct a single indicator function, we use $2m + 1$ activation functions, with depth 2 and width $2m$. If we restrict ourselves to ReLU neural network, we use $4m + 2$ neurons, with depth 4 and width $2m$; in contrast, Baader et al. (2020) used $10m - 3$ ReLu functions, with depth $3 + \log_2(m)$, and width $4m$.

## 7 COMPLETE PROOF CONSTRUCTION OF AUA THEOREM

We have shown how to approximate an indicator function and how to control the precision of its abstract interpretation (Thm. 6.2). We now complete the construction of the neural network $N$ following the technique of Baader et al. (2020) for ReLU networks. Because we use an arbitrary squashable function to approximate the sign function, this introduces extra imprecision in comparison with ReLUs. We thus need a finer function slicing to accommodate it, i.e., we use a slicing size of $\delta/3$ instead of $\delta/2$ in Baader et al. (2020). We provide the detailed analysis in the appendix. In what follows, we outline on how to build the network $N$ that satisfies the AUA theorem.

**Slicing $f$.** Let $f : C \to \mathbb{R}$ be the continuous function we need to approximate, and $\delta$ be the approximation tolerance, as per AUA theorem statement (Thm. 4.2). Assume $\min f(C) = 0$.[5] Let $u = \max f(C)$. In other words, the range of $f$ is $[0, u]$.

Let $\tau = \frac{\delta}{3}$. We will decompose $f$ into a sequence of *function slices* $f_i$, whose values are restricted to $[0, \tau]$. Let $K = \lfloor u/\tau \rfloor$. The sum of the sequence of function slices is $f$. The sequence of functions $f_i : C \to [0, \tau]$, for $i \in \{0, \ldots, K\}$, is:

$$f_i(\mathbf{x}) = \begin{cases} f(\mathbf{x}) - i\tau, & i\tau < f(\mathbf{x}) \leqslant (i+1)\tau \\ 0, & f(\mathbf{x}) \leqslant i\tau \\ \tau, & (i+1)\tau < f(\mathbf{x}) \end{cases}$$

**Approximating $f_i$.** We will use the indicator approximation $N_G$ (Eq. (3)) to construct a neural network $N_i$ that approximates $f_i$. Let $\mathcal{G}$ be the set of boxes whose vertices are in the grid. Because $C$ is compact, $|\mathcal{G}|$ is finite. Consider $\frac{1}{\tau} f_i(\mathbf{x})$; it is roughly similar to an indicator function for the set $S = \{\mathbf{x} \in C \mid f(\mathbf{x}) > (i+1)\tau\}$, i.e., indicating when $f(\mathbf{x})$ is greater than the upper bound of the $i$th slice. To approximate $\frac{1}{\tau} f_i(\mathbf{x})$, we will consider all boxes in $\mathcal{G}$ that are subsets of $S$, and construct an indicator function to tell us whether an input $\mathbf{x}$ is in those boxes. Let $\mathcal{G}_i = \{G \in \mathcal{G} \mid f(G) > (i+1)\tau\}$.

Now construct $N_i(\mathbf{x})$ that approximates $\frac{1}{\tau} f_i(\mathbf{x})$ as $N_i(\mathbf{x}) = t\left(\mu\left(\sum_{G \in \mathcal{G}_i} N_G(\mathbf{x}) - 0.5\right)\right)$.

**Sum all $N_i$.** Because $\sum_{i=0}^{K} f_i(\mathbf{x}) = f(\mathbf{x})$, and $N_i(\mathbf{x})$ approximates $\frac{1}{\tau} f_i(\mathbf{x})$, we will construct the neural network $N$ as $N(\mathbf{x}) = \tau \sum_{i=0}^{K} N_i(\mathbf{x})$.

$N$ $\delta$-abstractly approximates $f$; therefore, the AUA theorem holds.

---

[5]Otherwise, we can shift $f$ such that $\min f(C) = 0$.

## 8 CONCLUSION

We have shown that the AUA theorem holds for most practical neural networks, and demonstrated that in theory interval analysis can certify the robustness of neural networks. In the future, one might be interested in reducing the size of the neural network constructed in course of our proof, for example, by allowing the use of richer domains, like zonotopes and polyhedra.

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

## A  VECTOR-VALUED NETWORKS AND ROBUSTNESS

In this section, we extend the AUA theorem to vector-valued functions. We also extend our robustness results to $n$-ary classifiers.

### A.1  HIGHER-DIMENSIONAL FUNCTIONS

**Vector-valued neural networks.** So far we have considered scalar-valued neural networks. We can generalize the neural-network grammar (Def. 3.1) to enable vector-valued neural networks. Simply, we can compose a sequence of $n$ scalar-valued neural networks to construct a neural network whose range is $\mathbb{R}^n$. Formally, we extend the grammar as follows, where $E_i$ are the scalar-valued sub-neural networks.

**Definition A.1** (Vector-valued neural network grammar). *A neural network $N : \mathbb{R}^m \to \mathbb{R}^n$ is defined as follows*

$$
\begin{aligned}
N &:\!- \quad (E_1, \ldots, E_n) \\
E &:\!- \quad c \\
&\quad\mid\quad x_i \\
&\quad\mid\quad E_1 + E_2 \\
&\quad\mid\quad c * E_2 \\
&\quad\mid\quad t(E_1, \ldots, E_k)
\end{aligned}
$$

*where $c \in \mathbb{R}$, $x_i$ is one of the $m$ inputs to the network, and $t$ is an activation function.*

**Example A.2.** *Consider the following neural network $N : \mathbb{R}^2 \to \mathbb{R}^2$:*

$$
N(\mathbf{x}) = (\sigma(x_1 + 0.5x_2), \ \sigma(0.1x_1 + 0.3x_2))
$$

*which we can pictorially depict as the following graph:*

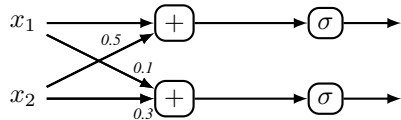

**Generalized AUA theorem.** We now generalize the AUA theorem to show that we can $\delta$-abstractly approximate vector-valued functions.

**Theorem A.3.** *Let* $f : C \to \mathbb{R}^n$ *be a continuous function with compact domain* $C \subset \mathbb{R}^m$*. Let* $\delta > 0$*. Then, there exists a neural network* $N : \mathbb{R}^n \to \mathbb{R}^m$ *such that for every box* $B \subseteq C$*, and for all* $i \in [1, m]$,

$$[l_i + \delta, u_i - \delta] \subseteq N^\#(B)_i \subseteq [l_i - \delta, u_i + \delta] \tag{4}$$

*where*

1. $N^\#(B)_i$ *is the ith interval in the box* $N^\#(B)$*, and*

2. $l_i = \min S_i$ *and* $u_i = \max S_i$*, where* $S = f(B)$ *(recall that* $S_i$ *is the set of ith element of every vector in S).*

*Proof.* From the AUA theorem, we know that there exists a neural network $N_i$ that $\delta$-abstractly approximates $f_i : C \to \mathbb{R}$, which is like $f$ but only returns the $i$th output. We can then construct the network $N = (N_1, \ldots, N_n)$. Since each $N_i$ satisfies Eq. (4) separately, then $N$ $\delta$-abstractly approximates $f$. $\square$

## A.2 ROBUSTNESS IN $n$-ARY CLASSIFICATION

We now extend the definition of $\epsilon$-robustness to $n$-ary classifiers. We use a function $f : C \to \mathbb{R}^n$ to denote an $n$-class classifier. $f$ returns a value for each of the $n$ classes; the class with the largest value is the result of classification. We assume there are no ties. Formally, for a given $\mathbf{x} \in C$, we denote classification by $f$ as $\text{class}(f(\mathbf{x}))$, where

$$\text{class}(\mathbf{y}) = \underset{i \in \{1, \ldots, m\}}{\arg\max} \; y_i$$

**Definition A.4** ($n$-ary robustness). *Let* $M \subset C$*. We say that* $f$ *is* $\epsilon$-robust *on* $M$*, where* $\epsilon > 0$*, iff for all* $\mathbf{x} \in M$ *and* $\mathbf{x}' \in R_\epsilon(\mathbf{x})$*, we have* $\text{class}(f(\mathbf{x})) = \text{class}(f(\mathbf{x}'))$.

We now extend the certifiably robust neural networks definition to the $n$-class case. Recall that $R_\epsilon(\mathbf{x}) = \{\mathbf{x}' \mid ||\mathbf{x} - \mathbf{x}'|| \leqslant \epsilon\}$.

**Definition A.5** (Certifibly robust networks). *A neural network* $N$ *is* $\epsilon$-certifiably robust *on* $M$ *iff, for all* $\mathbf{x} \in M$*, for all* $\mathbf{y}, \mathbf{y}' \in N^\#(R_\epsilon(\mathbf{x}))$*, we have* $\text{class}(\mathbf{y}) = \text{class}(\mathbf{y}')$.

**Existence of robust networks.** We now show existence of robust networks that approximate some robust $n$-ary classifier $f$.

**Theorem A.6** (Existence of robust networks). *Let* $f : C \to \mathbb{R}^n$ *be a continuous function that is* $\epsilon$-robust on set $M$*. Then, there exists a neural network that*

1. *agrees with* $f$ *on* $M$*, i.e.,* $\forall \mathbf{x} \in M. \, \text{class}(N(\mathbf{x})) = \text{class}(f(\mathbf{x}))$*, and*

2. *is* $\epsilon$-provably robust on $M$*.*

*Proof.* First, we need to post-process the results of $f$ as follows: For all $\mathbf{x} \in C$,

$$\hat{f}(\mathbf{x}) = (0, \ldots, |y_i|, \ldots, 0)$$

where $\mathbf{y} = f(\mathbf{x})$ and $\text{class}(f(\mathbf{x})) = i$. In other words, $\hat{f}$ is just like $f$, but it zeroes out the values of all but the output class $i$. This is needed since the interval domain is non-relational, and therefore it cannot capture relations between values of different classes, namely, keeping track which one is larger. Note that if $f$ is continuous, then $\hat{f}$ is continuous.

Let $\delta'$ be the smallest non-zero element of any vector in the set $\{\hat{f}(\mathbf{x}) \mid \mathbf{x} \in C\}$. Following the AUA theorem, let $N$ be a neural network that $\delta$-abstractly approximates $\hat{f}$, where $\delta < 0.5\delta'$.

STATEMENT (1): Pick any $\mathbf{x} \in M$. Let the $i$th element of $\hat{f}(\mathbf{x}) \neq 0$; call it $c$. By construction $i = \text{class}(f(\mathbf{x}))$. Let $N(\mathbf{x}) = (y_1, \ldots, y_n)$. By AUA theorem, we know that $0 \leqslant y_j < 0.5\delta'$, for $j \neq i$, and $y_i \geqslant c - 0.5\delta'$. Since $c \geqslant \delta'$, $\text{class}(N(\mathbf{x})) = \text{class}(f(\mathbf{x})) = i$.

STATEMENT (2): Let $\mathbf{x} \in M$. Let $S = \hat{f}(R_\epsilon(\mathbf{x}))$. Let $S_i$ be the projection of all vectors in $S$ on their $i$th element, where $i = \text{class}(\hat{f}(\mathbf{x}))$. We know that $\min S_i \geqslant \delta'$. $\min S_i$ exists because $R_\epsilon(\mathbf{x})$ is compact, so are $S$ and $S_i$. By construction of $\hat{f}$ and the fact that $f$ is robust, all other elements of vectors of $S$ are zero, i.e., $S_j = \{0\}$, for $j \neq i$.

Let $N^\#(R_\epsilon(\mathbf{x})) = [l_j, u_j]$. By AUA theorem and its proof, for $j \neq i$, we have $[l_j, u_j] \subset [0, 0.5\delta']$. Similarly, $[l_i, u_i] \subseteq [\min S_i - 0.5\delta', u_i] \subseteq [0.5\delta', u_i]$. It follows that for all $\mathbf{y}, \mathbf{y}' \in N^\#(R_\epsilon(\mathbf{x}))$, we have $\text{class}(\mathbf{y}) = \text{class}(\mathbf{y}') = i$. This is because any value in $[\delta' - 0.5\delta', u_i]$ is larger than any value in $[0, 0.5\delta')$.

**Notice that Thm. 4.5 is a special case, so it also holds.**

$\square$

# B  APPENDIX: ELIDED PROOFS

## B.1  PROOF OF PROPOSITION 3.3

It is easy to see that all the activation functions $t$ are monotonically increasing with

$$\lim_{x \to -\infty} t(x) = l \quad \text{and} \quad \lim_{x \to \infty} t(x) = \infty.$$

for some $l \in \mathbb{R}$.

Because $t$ is increasing, $t(-x)$ and $t(1 - x)$ are both decreasing; thus, their composition $t(1 - t(-x))$ is increasing.

$$\lim_{x \to -\infty} t(1 - t(-x)) = t(\lim_{x \to -\infty} (1 - t(-x))) = l$$

$$\lim_{x \to \infty} t(1 - t(-x)) = t(1 - \lim_{x \to \infty} t(-x)) = t(1 - l)$$

**ReLU.:** $l = 0$, and $t(1 - l) = \text{ReLU}(1 - 0) = 1$.

**ELU.:** $l = -1$, and $t(1 - l) = \text{ELU}(2) = 2$.

**softplus.:** $l = 0$, and $t(1 - l) = \text{softplus}(1) = \log(1 + e)$.

**smoothReLU.:** $l = 0$, and $t(1 - l) = \text{smoothReLU}_a(1) = 1 - \frac{1}{a}\log(a + 1)$. (One can easily verify that $\frac{1}{a}\log(a + 1) < 1$ for $a \neq 0$).

## B.2  CHOICE OF PARAMETERS $\theta$ AND $\epsilon$

Because the our construction works for any fixed $\theta$ and $\epsilon$, we will choose $\theta = \min(\frac{1}{K+1}, \frac{1}{4m+2}, \frac{1}{4|\mathcal{G}|})$, where $\tau$, $K$ and $\mathcal{G}$ are defined in Sec. 7; and $\epsilon < 0.5$ be such that if $\|\mathbf{x} - \mathbf{y}\|_\infty \leqslant \epsilon$, then $|f(\mathbf{x}) - f(\mathbf{y})| < \tau$. The latter is achievable from the Heine–Cantor Theorem (see Rudin (1986)), so $f$ is uniformly continuous on $C$.

## B.3  PROPERTIES OF $\hat{t}_i$

The following lemmas show that $\hat{t}_i$ roughly behaves like an indicator function: its value within a box's $i$th dimension $[a_i, b_i]$ is $\approx 1$; its value outside of the neighborhood is $\approx 0$; its value globally is bounded by 1. We will analyze the values of the two terms in $\hat{t}_i$.

The following lemma states that if $x$ is within the box's $i$th dimension, then the first term is close to 1 and the second term is close to 0, resulting in $\hat{t}_i(x) \approx 1$.

**Lemma B.1.** *If $x \in [a_i, b_i]$, then the following is true:*

1. $t(\mu(x + 0.5\epsilon - a_i)) \in (1 - \theta, 1]$.

2. $t(\mu(x - 0.5\epsilon - b_i)) \in [0, \theta)$.

*Proof.* STATEMENT (1): Because $x \geqslant a_i$, $x + 0.5\epsilon - a_i \geqslant 0.5\epsilon$. From Lem. 6.1, $t(\mu(x + 0.5\epsilon - a_i)) \in (1 - \theta, 1]$.

STATEMENT (2): Because $x \leqslant b_i$, $x - 0.5\epsilon - b_i \leqslant -0.5\epsilon$. From Lem. 6.1, $t(\mu(x - 0.5\epsilon - b_i)) \in [0, \theta)$.

$\square$

The next two lemmas state that if $x$ is outside the neighborhood, then the two terms are similar, resulting in a $\hat{t}_i(x) \approx 0$.

**Lemma B.2.** *If $x \leqslant a_i - \epsilon$, then the following is true:*

1. $t(\mu(x + 0.5\epsilon - a_i)) \in [0, \theta)$.

2. $t(\mu(x - 0.5\epsilon - b_i)) \in [0, \theta)$.

*Proof.* STATEMENT (1): Because $x \leqslant a_i - \epsilon$, $x + 0.5\epsilon - a_i \leqslant -0.5\epsilon$. From Lem. 6.1, $t(\mu(x + 0.5\epsilon - a_i)) \in [0, \theta)$.

STATEMENT (2): Because $x \leqslant a_i - \epsilon$ and $a_i < b_i$, $x \leqslant b_i - \epsilon$. Then $x - 0.5\epsilon - b_i \leqslant -0.5\epsilon$. From Lem. 6.1, $t(\mu(x - 0.5\epsilon - b_i)) \in [0, \theta)$.

$\square$

**Lemma B.3.** *If $x \geqslant b_i + \epsilon$, then the following is true:*

1. $t(\mu(x + 0.5\epsilon - a_i)) \in (1 - \theta, 1]$.

2. $t(\mu(x - 0.5\epsilon - b_i)) \in (1 - \theta, 1]$.

*Proof.* STATEMENT (1): Because $x \geqslant b_i + \epsilon$ and $b_i \geqslant a_i$, $x \geqslant a_i + \epsilon$. Then $x + 0.5\epsilon - a_i \geqslant 0.5\epsilon$. From Lem. 6.1, $t(\mu(x + 0.5\epsilon - a_i)) \in (1 - \theta, 1]$.

STATEMENT (2): Because $x \geqslant b_i + \epsilon$, $x - 0.5\epsilon - b_i \geqslant 0.5\epsilon$. From Lem. 6.1, $t(\mu(x - 0.5\epsilon - b_i)) \in (1 - \theta, 1]$.

$\square$

### B.3.1 ABSTRACT PRECISION OF $\hat{t}_i$

We are now ready to prove properties about the abstract interpretation of our 1-dimensional indicator approximation, $\hat{t}_i$. The following lemma states that the abstract interpretation of $\hat{t}_i$, $\hat{t}_i^\#(B)$, is quite precise: if the 1-dimensional input box $B$ is outside the neighborhood of $G$, on $G$'s $i$th dimension, then the output box is within $\theta$ from 0; if the input box $B$ is within the $i$th dimension of $G$, then the output box is within $2\theta$ from 1.

**Lemma B.4** (Abstract interpretation of $\hat{t}_i$). *For a 1-dimensional box $B$, the following is true:*

1. $\hat{t}_i^\#(B) \subset (-\infty, 1]$.

2. *If $B \subseteq (-\infty, a_i - \epsilon]$ or $B \subseteq [b_i + \epsilon, \infty)$, then $\hat{t}_i^\#(B) \subseteq (-\theta, \theta)$.*

3. *If $B \subseteq [a_i, b_i]$, then $\hat{t}_i^\#(B) \subseteq (1 - 2\theta, 1]$.*

*Proof.* We begin the proof by simplifying the expression $\hat{t}_i^{\#}(B)$. Recall that $\hat{t}(x) = t(\mu(x + 0.5\epsilon - a_i)) - t(\mu(x - 0.5\epsilon - b_i))$. Let $B = [a, b]$. By applying abstract transformer $t^{\#}$ (Def. 3.6) and subtracting the two terms, we get $\hat{t}_i^{\#}(B) = [T_1 - T_4, T_2 - T_3]$, where

$$T_1 = t(\mu(a + 0.5\epsilon - a_i)) \qquad T_2 = t(\mu(b + 0.5\epsilon - a_i))$$
$$T_3 = t(\mu(a - 0.5\epsilon - b_i)) \qquad T_4 = t(\mu(b - 0.5\epsilon - b_i))$$

We are now ready to prove the three statements.

STATEMENT (1): By the definition of $t$, $\forall x. \, t(x) \in [0, 1]$, so $T_1, T_2, T_3, T_4 \in [0, 1]$. Therefore, the upper bound of $\hat{t}^{\#}(B)$ is $T_2 - T_3 \leqslant 1$.

STATEMENT (2):

*Case 1:* $B \subseteq (-\infty, a_i - \epsilon]$. From Lem. B.2, $T_1, T_2, T_3, T_4 \in [0, \theta)$, then $T_2 - T_3 < \theta$, and $T_1 - T_4 > -\theta$.

*Case 2:* $B \subseteq [b_i + \epsilon, \infty)$. From Lem. B.3, $T_1, T_2, T_3, T_4 \in (1 - \theta, 1]$, then $T_2 - T_3 < \theta$, and $T_1 - T_4 > -\theta$.

In either case, $\hat{t}_i^{\#}(B) \subseteq (-\theta, \theta)$.

STATEMENT (3): If $B \subseteq [a_i, b_i]$, $a, b \in [a_i, b_i]$. From Lem. B.1(1), $T_1, T_2 \in (1 - \theta, 1]$. From Lem. B.1(2), $T_3, T_4 \in [0, \theta)$. Then $T_1 - T_4 > 1 - 2\theta$ and $T_2 - T_3 \leqslant 1$.

Therefore, $\hat{t}_i^{\#}(B) \subseteq (1 - 2\theta, 1]$. □

### B.4 ABSTRACT PRECISION OF $N_G$

We are now ready to analyze the abstract precision of $N_G$. We first consider $H_i$ in the following lemma. For any box $B \subseteq C$, let $B_i$ be its projection on dimension $i$, which is an interval.

The following lemma states that if $B$ is in the box $G$, then $\sum_i H_i^{\#}$ is positive; otherwise, if $B$ is outside the neighborhood of $G$, then $\sum_i H_i^{\#}$ is negative.

**Lemma B.5** (Abstract interpretation of $H_i$). *For any box $B \subseteq C$, the following is true:*

1. *If $B \subseteq G$, then $\sum_{i=1}^{m} H_i^{\#}(B_i) \subseteq (0, \infty)$.*

2. *If $B \subseteq C \setminus \nu(G)$, then $\sum_{i=1}^{m} H_i^{\#}(B_i) \subseteq (-\infty, -\epsilon)$.*

*Proof.* STATEMENT (1): If $B \subseteq G$, then $\forall i. \, B_i \subseteq [a_i, b_i]$. From Lem. B.4 (3), $\hat{t}_i^{\#}(B_i) \subseteq (1 - 2\theta, 1]$; thus,

$$\begin{aligned} H_i^{\#}(B_i) &= \hat{t}_i^{\#}(B_i) +^{\#} -(1 - 2\theta)^{\#} \\ &\subseteq (0, 2\theta] \\ &\subset (0, \infty) \end{aligned}$$

Sum over all $m$ dimensions, $\sum_{i=1}^{m} H_i^{\#}(B_i) \subseteq \sum_{i=1}^{m} (0, \infty) = (0, \infty)$.

STATEMENT (2): If $B \subseteq C \setminus \nu(G)$, then there is a dimension $j$ such that either $B_j \subseteq (-\infty, a_j - \epsilon]$ or $B_j \subseteq [b_j + \epsilon, \infty)$. From Lem. B.4 (2), we know that $\hat{t}^{\#}(B_j) \subseteq (-\theta, \theta)$. Therefore,

$$\begin{aligned} H_j^{\#}(B_j) &= \hat{t}^{\#}(B_j) +^{\#} -(1 - 2\theta)^{\#} \\ &\subseteq (\theta - 1, 3\theta - 1) \end{aligned} \tag{5}$$

For the remaining $m - 1$ dimensions, from Lem. B.4 (1), we know that $\hat{t}^{\#}(B_i) \subset (-\infty, 1]$ when $i \neq j$. Therefore,

$$\begin{aligned} H_i^{\#}(B_i) &= \hat{t}^{\#}(B_i) +^{\#} -(1 - 2\theta)^{\#} \\ &\subseteq (-\infty, 2\theta] \end{aligned} \tag{6}$$

Take the sum of all the $m - 1$ dimensions,

$$
\begin{aligned}
\sum_{i \in \{1,\dots,m\} \backslash \{j\}} H_i^\#(B_i) &\subseteq \sum_{i \in \{1,\dots,m\} \backslash \{j\}} (-\infty, 2\theta] &&\text{(substitute Eq. (6))} \\
&= [m-1, m-1] *^\# (-\infty, 2\theta] &&\text{(turn sum into } *^\#) \qquad (7) \\
&= (-\infty, 2(m-1)\theta] &&\text{(apply } *^\#)
\end{aligned}
$$

Now, take sum over all the $m$ dimensions,

$$
\begin{aligned}
\sum_{i=1}^m H_i^\#(B_i) &= \sum_{i \in \{1,\dots,m\} \backslash \{j\}} H_i^\#(B_i) +^\# H_j^\#(B_j) &&\text{(decompose sum)} \\
&\subseteq (-\infty, 2(m-1)\theta] +^\# (\theta - 1, 3\theta - 1) &&\text{(substitute Eqs. (5) and (7))} \\
&= (-\infty, (2m+1)\theta - 1) &&\text{(apply } *^\#)
\end{aligned}
$$

Because of our choice of $\theta$, $\theta \leqslant \frac{1}{4m+2}$ (see Appendix B.2). Then $(2m+1)\theta \leqslant \frac{2m+1}{4m+2} = 0.5$, and therefore

$$
\sum_{i=1}^m H_i^\#(B_i) \subseteq (-\infty, -0.5)
$$

Also we have assumed that $\epsilon < 0.5$ (see Appendix B.2); therefore

$$
\sum_{i=1}^m H_i^\#(B_i) \subseteq (-\infty, -\epsilon)
$$

$\square$

### B.4.1  PROOF OF THM. 6.2

*Proof.*
STATEMENT (1): See definition of $N_G$ in Eq. (3). The outer function of $N_G$ is $t$, whose range is $[0, 1]$ by the definition of squashable functions and our assumption that the left and right limits are $0$ and $1$. Therefore, $N_G^\#(B) \subseteq [0, 1]$.

STATEMENT (2): If $B \subseteq G$, from Lem. B.5, we know that $\sum_{i=1}^m H_i^\#(B_i) \subseteq (0, \infty)$. Then,

$$
\sum_i^m H_i^\#(B_i) +^\# (0.5\epsilon)^\# \quad \subseteq \quad (0, \infty) +^\# (0.5\epsilon)^\# \quad \subseteq \quad (0.5\epsilon, \infty)
$$

From Lem. 6.1, we know that if $x \geqslant 0.5\epsilon$, then $1 - \theta < t(\mu x) \leqslant 1$. Therefore,

$$
N_G^\#(B) = t^\#(\mu^\# *^\# (0.5\epsilon, \infty)) \quad \subseteq \quad (1 - \theta, 1]
$$

STATEMENT (3): If $B \subseteq C \backslash \nu(G)$, from Lem. B.5, we know that $\sum_{i=1}^m H_i^\#(B_i) \subseteq (-\infty, -\epsilon)$. Then,

$$
\sum_{i=1}^m H_i^\#(B_i) +^\# (0.5\epsilon)^\# \quad \subseteq \quad (-\infty, -\epsilon) +^\# (0.5\epsilon)^\# \quad \subseteq \quad (-\infty, -0.5\epsilon)
$$

From Lem. 6.1, we know that if $x \leqslant -0.5\epsilon$, then $0 \leqslant t(\mu x) < \theta$. Therefore,

$$
N_G^\#(B) = t^\#(\mu^\# *^\# (-\infty, -0.5\epsilon)) \quad \subseteq \quad [0, \theta)
$$

$\square$

### B.5  ABSTRACT INTERPRETATION OF $N_i$

Observe how for any box $B \subseteq C$ from the abstract domain, it is overapproximated by a larger box $G \supseteq B$ from the finitely many boxes in the $\epsilon$-grid. Intuitively, our abstract approximation of $N_i$ incurs an error when the input $B$ is not in the grid. We formalize this idea by extending the notion of neighborhood (Sec. 6) to boxes from the abstract domain. For a box $B \subseteq C$, if $B \in \mathcal{G}$, then $B$'s neighborhood $G_B = \nu(B)$; otherwise, let $G_B$ be the smallest $G \in \mathcal{G}$, by volume, such that $B \subseteq G$. Note that $G_B$ is uniquely defined.

The following lemma says that considering the neighborhood of $B$ only adds up to $\tau$ of imprecision to the collecting semantics of $f$.

**Lemma B.6** (Properties of $G_B$). *The following is true:*

1. *If $f(B) \geqslant \beta$, then $f(G_B) \geqslant \beta - \tau$.*

2. *If $f(B) \leqslant \beta$, then $f(G_B) \leqslant \beta + \tau$.*

*Proof.* Both of the statements follow from our choice of $\epsilon$ in constructing the grid (see Appendix B.2). If $\|\mathbf{x} - \mathbf{x}\|_\infty \leqslant \epsilon$, then $|f(\mathbf{x}) - f(\mathbf{y})| < \tau$. Consider the $B$ and its neighborhood $G_B$. By definition of neighborhood, $\forall \mathbf{x} \in G_B, \exists \mathbf{y} \in B$, such that $\|\mathbf{x} - \mathbf{y}\|_\infty \leqslant \epsilon$.

STATEMENT (1) Because $f(B) \geqslant \beta$, then $f(\mathbf{y}) \geqslant \beta$, so $f(\mathbf{x}) \geqslant f(\mathbf{y}) - \tau \geqslant \beta - \tau$. Then $\forall \mathbf{x} \in G_B$, $f(\mathbf{x}) \geqslant \beta - \tau$.

STATEMENT (2) Because $f(B) \leqslant \beta$, then $f(\mathbf{y}) \leqslant \beta$, so $f(\mathbf{x}) \leqslant f(\mathbf{y}) + \tau \leqslant \beta + \tau$. Then $\forall \mathbf{x} \in G_B$, $f(\mathbf{x}) \leqslant \beta + \tau$. $\square$

**Theorem B.7** (Abstract interpretation of $N_i$). *For any box $B \subseteq C$, let $u = \max f(B)$, and $l = \min f(B)$. The following is true:*

1. $N_i^\#(B) \subseteq [0, 1]$.

2. *If $l \geqslant (i + 2)\tau$, then $\exists u_i \in (1 - \theta, 1]$ such that $[u_i, u_i] \subseteq N_i^\#(B) \subseteq (1 - \theta, 1]$.*

3. *If $u \leqslant (i - 1)\tau$, then $\exists l_i \in [0, \theta)$ such that $[l_i, l_i] \subseteq N_i^\#(B) \subseteq [0, \theta)$.*

*Proof.* We begin by noting that in Statement (2), $[u_i, u_i] \subseteq N_i^\#(B)$ for some $u_i \in (1 - \theta, 1]$ is a direct corollary of $N_i^\#(B) \subseteq (1 - \theta, 1]$. Because if $N_i^\#(B) \subseteq (1 - \theta, 1]$, and $N_i^\#(B) \neq \emptyset$, then $N_i^\#(B)$ contains at least one point in $(1 - \theta, 1]$. Similarly, in Statement (3), $[l_i, l_i] \subseteq N_i^\#(B)$ for some $l_i \in [0, \theta)$ is a direct corollary of $N_i^\#(B) \subseteq [0, \theta)$.

In Appendix B.2, we have chosen that $\theta \leqslant \frac{1}{4|\mathcal{G}|}$, a fact we will use later in the proof.

STATEMENT (1): The outer function of $N_i$ is $t$, whose range is $[0, 1]$, by the definition of squashable function and our construction, so $N_i^\#(B) \subseteq [0, 1]$.

STATEMENT (2): Because $f(B) \geqslant (i + 2)\tau$, by Lem. B.6, $f(G_B) \geqslant (i + 1)\tau$, so $G_B \in \mathcal{G}_i$. Thus, we can break up the sum as follows:

$$\sum_{G \in \mathcal{G}_i} N_G(\mathbf{x}) = \left( \sum_{G \in (\mathcal{G}_i \setminus \{G_B\})} N_G(\mathbf{x}) \right) + N_{G_B}(\mathbf{x})$$

From Thm. 6.2, $N_{G_B}^\#(B) \subseteq (1 - \theta, 1]$, and $N_G^\#(B) \subseteq [0, 1]$ for $G \in \mathcal{G}_i \setminus \{G_B\}$. Therefore, we can conclude the following two facts:

$$\sum_{G \in \mathcal{G}_i} N_G^\#(B) \subseteq (1-\theta, \infty) \quad \text{and} \quad \sum_{G \in \mathcal{G}_i} N_G^\#(B) +^\# [-0.5, -0.5] \subseteq (0.5 - \theta, \infty) \subset (0.5\epsilon, \infty)$$

The second inequality follows from the fact that we assumed $\theta \leqslant \frac{1}{4|\mathcal{G}|} \leqslant 0.25$ (above) and $\epsilon < 0.5$ (see Appendix B.2). Therefore, $0.5 - \theta > 0.25 > 0.5\epsilon$.

It follows from Lem. 6.1 that

$$N_i^\#(B) = t^\# \left( \mu^\# *^\# \left( \sum_{G \in \mathcal{G}_i} N_G^\#(B) +^\# [-0.5, -0.5] \right) \right) \subseteq (1 - \theta, 1]$$

STATEMENT (3): If $u \leqslant (i - 1)\tau$, we will show that $\forall G \in \mathcal{G}_i . B \subset C \setminus \nu(G)$.

Pick any $G \in \mathcal{G}_i$, then we have $f(G) \geqslant (i+1)\tau$. Thus, from Lem. B.6, $f(G_B) \geqslant i\tau$. Recall that if $B \in \mathcal{G}$, then $G_B = \nu(B)$. Hence, $f(\nu(G)) \geqslant i\tau$. However, $f(B) \leqslant u \leqslant (i-1)\tau$, so $B \cap \nu(G) = \emptyset$. Equivalently, $B \subset C \setminus \nu(G)$.

From Thm. 6.2, $\forall G \in \mathcal{G}_i. N_G^{\#}(B) \subseteq [0, \theta)$, so

$$\sum_{G \in \mathcal{G}_i} N_G^{\#}(B) \subseteq [0, |\mathcal{G}_i|\theta) \subseteq [0, |\mathcal{G}|\theta)$$

We assumed that $\theta \leqslant \frac{1}{4|\mathcal{G}|}$ and $\epsilon < 0.5$ (see Appendix B.2), so $|\mathcal{G}|\theta \leqslant 0.25$, and $|\mathcal{G}|\theta - 0.5 \leqslant -0.25 \leqslant -0.5\epsilon$. Hence,

$$\sum_{G \in \mathcal{G}_i} N_G^{\#}(B) \subseteq [0, 0.25) \quad \text{and} \quad \sum_{G \in \mathcal{G}_i} N_G^{\#}(B) +^{\#} [-0.5, -0.5] \subseteq [-0.5, -0.25) \subseteq (-\infty, -0.5\epsilon)$$

It follows from Lem. 6.1 that

$$N_i^{\#}(B) = t^{\#} \left( \mu^{\#} *^{\#} \left( \sum_{G \in \mathcal{G}_i} N_G^{\#}(B) +^{\#} [-0.5, -0.5] \right) \right) \subseteq [0, \theta)$$

$\square$

## B.6 Abstract interpretation of $N$

Before proceeding with the proof, we give a general lemma that will be useful in our analysis. The lemma follows from the fact that, by construction, $\theta \leqslant \frac{1}{K+1}$.

**Lemma B.8.** *If $\eta_0, \ldots, \eta_K \in [-\theta, \theta]$, then $\sum_{i=0}^{K} \eta_i \in [-1, 1]$.*

*Proof.* This simply follow from the choice of $\theta \leqslant \frac{1}{K+1}$. $\square$

**Proof outline of Thm. 4.2.** Our proof involves three pieces, outlined below:

(A) Because $N^{\#}(B) = \tau^{\#} *^{\#} \sum_{i=0}^{K} N_i^{\#}(B)$, we need only analyze $\sum_{i=0}^{K} N_i^{\#}(B)$. We will decompose the sum into five sums and analyze each separately, arriving at five results of the form:

$$\left[ \tilde{L}_{1j}, \tilde{U}_{1j} \right] \subseteq \sum_{i \in S_j} N_i^{\#}(B) \subseteq \left[ \tilde{L}_{2j}, \tilde{U}_{2j} \right]$$

for $j \in \{1, \ldots, 5\}$, where $\bigcup_j S_j = \{0, \ldots, K\}$ and $S_j$ are mutually disjoint sets.

(B) Then, we sum over all five cases, getting

$$\left[ \sum_{j=1}^{5} \tilde{L}_{1j}, \sum_{j=1}^{5} \tilde{U}_{1j} \right] \subseteq \sum_{i=0}^{K} N_i^{\#}(B) \subseteq \left[ \sum_{j=1}^{5} \tilde{L}_{2j}, \sum_{j=1}^{5} \tilde{U}_{2j} \right]$$

(C) Let $L_i = \tau \sum_{j=1}^{5} \tilde{L}_{ij}$ and $U_i = \tau \sum_{j=1}^{5} \tilde{U}_{ij}$. Then, we get the bound $[L_1, U_1] \subseteq N^{\#}(B) \subseteq [L_2, U_2]$.

Finally, we show that $[L_2, U_2] \subseteq [l - \delta, u + \delta]$ and $[l + \delta, u - \delta] \subseteq [L_1, U_1]$. Equivalently, we will show that

$$l - \delta \leqslant L_2 \leqslant L_1 \leqslant l + \delta \quad \text{and} \quad u - \delta \leqslant U_1 \leqslant U_2 \leqslant u + \delta$$

**Proof assumptions.** We will assume that $l \in [p\tau, (p+1)\tau)$ and $u \in [q\tau, (q+1)\tau)$, for some $p \leqslant q \leqslant K$. Additionally, let $c, d \in B$ be such that $f(c) = l$ and $f(d) = u$.

**Step A: Decompose sum and analyze separately.** We begin by decomposing the sum into five terms.

This is the most important step of the proof. We want to show that most $N_i$'s in $\sum_{i=0}^{K} N_i^{\#}(B)$ are (almost) precise. By almost we mean that their values are $\approx 1$ and $\approx 0$. The motivation is then to extract as many precise terms as possible. The only tool used in the analysis is Thm. B.7.

- Consider the function slices represented by Term 1 and 5; for example, Term 1 represents abstractions $N_i^{\#}$ of function slices $f_i$, for $i \in [0, p-2]$. The function slices of Term 1 and 5 are referred to in Thm. B.7 (Statements 2 and 3): they have an (almost) precise abstract interpretation. That is, the abstract semantics of $N_i^{\#}(B)$ and the collecting semantics of $f_i(B)$ agree. For Term 1, the abstract interpretation of all $N_i^{\#}(B) \approx [1, 1]$ and $f_i(B) = [\tau, \tau]$. For Term 5, the abstract interpretation of all $N_i^{\#}(B) \approx [0, 0]$ and $f_i(B) = [0, 0]$.

- Now consider function slices $f_i$, where $i \in [p+2, q-2]$. The abstraction of these function slices is also (almost) precise. We can see $f(c) = l$ is below the lower bound of the slices and $f(d) = u$ is above the upper bound of the slices. Hence, $f_i(d) = \tau$ and $N_i^{\#}(\{d\}) \approx [1, 1]$. Similarly, $f_i(c) = 0$ and $N_i^{\#}(\{c\}) \approx [0, 0]$. Because $c, d \in B$, and due to continuity of $f$, we have $f_i(B) = [0, 1]$, and $N_i^{\#}(B) \approx [0, 1]$.

- The remaining function slices are those in Term 2 and Term 4, and they are at the neighborhood of the boundary of $[l, u]$. Most precision loss of $N_i^{\#}(B)$ comes from those two terms.

This drives us to decompose the sum as follows:

$$\sum_{i=0}^{K} N_i^{\#}(B) \quad = \quad \underbrace{\sum_{i=0}^{p-2} N_i^{\#}(B)}_{\text{Term 1}} +^{\#} \underbrace{\sum_{i=p-1}^{p+1} N_i^{\#}(B)}_{\text{Term 2}} +^{\#} \underbrace{\sum_{i=p+2}^{q-2} N_i^{\#}(B)}_{\text{Term 3}} +^{\#} \underbrace{\sum_{i=q-1}^{q+1} N_i^{\#}(B)}_{\text{Term 4}} +^{\#} \underbrace{\sum_{i=q+2}^{K} N_i^{\#}(B)}_{\text{Term 5}}$$
(8)

We will analyze the five terms in Eq. (8) separately, and then take their sum to get the final result. For now, assume that $q \geqslant p + 3$; the $q \leqslant p + 2$ case will follow easily.

(i) Term 1: $\forall i \leqslant p - 2$, we have $p\tau \geqslant (i+2)\tau$. Because $l = \min f(B)$ and $l \in [p\tau, (p+1)\tau)$, then $f(B) \geqslant p\tau \geqslant (i+2)\tau$.

From Thm. B.7, $\exists u_i \in (1 - \theta, 1]$ such that $[u_i, u_i] \subseteq N_i^{\#}(B) \subseteq (1 - \theta, 1]$. Then $\sum_{i=0}^{p-2} [u_i, u_i] \subseteq \sum_{i=0}^{p-2} N_i^{\#}(B) \subseteq \sum_{i=0}^{p-2} (1 - \theta, 1]$.

$$\sum_{i=0}^{p-2} [u_i, u_i] \quad \subseteq \quad \sum_{i=0}^{p-2} N_i^{\#}(B) \quad \subseteq \quad (p-1)^{\#} *^{\#} (1 - \theta, 1]$$

(ii) Term 5: $\forall i \geqslant q + 2$, we have $(q+1)\tau \leqslant (i-1)\tau$. Because $u = \max f(B)$ and $u \in [q\tau, (q+1)\tau)$, then $f(B) < (q+1)\tau \leqslant (i-1)\tau$.

From Thm. B.7, $\exists l_i \in [0, \theta)$ such that $[l_i, l_i] \subseteq N_i^{\#}(B) \subseteq [0, \theta)$. Then $\sum_{i=q+2}^{K} [l_i, l_i] \subseteq \sum_{i=q+2}^{K} N_i^{\#}(B) \subseteq \sum_{i=q+2}^{K} [0, \theta)$.

$$\sum_{i=q+2}^{K} [l_i, l_i] \quad \subseteq \quad \sum_{i=q+2}^{K} N_i^{\#}(B) \quad \subseteq \quad (K - q - 1)^{\#} *^{\#} [0, \theta)$$

(iii) Term 3: $\forall i \in [p+2, q-2]$, we have $(p+1)\tau \leqslant (i-1)\tau$ and $q\tau \geqslant (i+2)\tau$.
$f(c) = l < (p+1)\tau \leqslant (i-1)\tau$, and $f(d) = u \geqslant q\tau \geqslant (i+2)\tau$.
From Thm. B.7, $N_i^{\#}(\{c\}) \subseteq [0, \theta)$ and $N_i^{\#}(\{d\}) \subseteq (1 - \theta, 1]$. Because $c, d \in B$, $[\theta, 1 - \theta] \subseteq N_i^{\#}(B)$.
Also by Thm. B.7, $N_i^{\#}(B) \subseteq [0, 1]$. Hence, $\sum_{i=p+2}^{q-2} [\theta, 1 - \theta] \subseteq \sum_{i=p+2}^{q-2} N_i^{\#}(B) \subseteq \sum_{i=p+2}^{q-2} [0, 1]$.

$$\sum_{i=p+2}^{q-2} [\theta, 1 - \theta] \quad \subseteq \quad \sum_{i=p+2}^{q-2} N_i^{\#}(B) \quad \subseteq \quad (q - p - 3)^{\#} *^{\#} [0, 1]$$

(iv) Term 2: $\forall i \in [p-1, p+1]$, since we have assumed that $q \geqslant p + 3$, then $q \geqslant p + 3 \geqslant i + 2$.

Because $f(d) \geqslant q\tau \geqslant (i+2)\tau$, from Thm. B.7, $\exists u_i \in (1-\theta, 1]$ such that $[u_i, u_i] \subseteq N_i^{\#}(\{d\}) \subseteq (1-\theta, 1]$.

Because $d \in B$, $[u_i, u_i] \subseteq N_i^{\#}(B)$. Hence, $[u_i, u_i] \subseteq N_i^{\#}(B) \subseteq [0, 1]$ and $\sum_{i=p-1}^{p+1} [u_i, u_i] \subseteq \sum_{i=p-1}^{p+1} N_i^{\#}(B) \subseteq \sum_{i=p-1}^{p+1} [0, 1]$.

$$\sum_{i=p-1}^{p+1} [u_i, u_i] \quad \subseteq \quad \sum_{i=p-1}^{p+1} N_i^{\#}(B) \quad \subseteq \quad 3^{\#} *^{\#} [0, 1]$$

(v) Term 4: For $\forall q - 1 \leqslant i \leqslant q+1$, because $q \geqslant p+3$, we have $p+1 \leqslant q-2 \leqslant i-1$. Then $f(c) = l < (p+1)\tau \leqslant (i-1)\tau$. From Thm. B.7, $\exists l_i \in [0, \theta)$ such that $[l_i, l_i] \subseteq N_i^{\#}(\{c\}) \subseteq [0, \theta)$.

Because $c \in B$, $[l_i, l_i] \subseteq N_i^{\#}(B)$. Thus, $[l_i, l_i] \subseteq N_i^{\#}(B) \subseteq [0, 1]$.

$$\sum_{i=q-1}^{q+1} [l_i, l_i] \subseteq \sum_{i=q-1}^{q+1} N_i^{\#}(B) \subseteq 3^{\#} *^{\#} [0, 1]$$

**Step B: Sum all five cases.** We now sum up all five inequalities we derived above to derive an overall bound of the sum in the form $[L_1', U_1'] \subseteq \sum_{i=0}^{K} N_i^{\#}(B) \subseteq [L_2', U_2']$. For example,

$$L_1' = \sum_{i=0}^{p-2} u_i + \sum_{i=q+2}^{K} l_i + \sum_{i=p+2}^{q-2} \theta + \sum_{i=p-1}^{p+1} u_i + \sum_{i=q-1}^{q+1} l_i$$

Recall that, by Thm. B.7, $\forall i \in \{0, \ldots, K\}$, $u_i \in (1-\theta, 1]$ and $l_i \in [0, \theta)$. Let $\tilde{l}_i = 1 - u_i$, so $\tilde{l}_i \in [0, \theta)$.

We simplify $L_1'$, $L_2'$, $U_1'$ and $U_2'$ as follows:

$$
\begin{aligned}
L_1' &= \sum_{i=0}^{p-2} u_i + \sum_{i=q+2}^{K} l_i + \sum_{i=p+2}^{q-2} \theta + \sum_{i=p-1}^{p+1} u_i + \sum_{i=q-1}^{q+1} l_i \\
&\qquad\qquad\qquad\qquad\qquad\qquad\qquad\qquad \text{(sum of the left bound)} \\
&= \sum_{i=0}^{p-2}(1 - \tilde{l}_i) + \sum_{i=q+2}^{K} l_i + \sum_{i=p+2}^{q-2} \theta + \sum_{i=p-1}^{p+1}(1 - \tilde{l}_i) + \sum_{i=q-1}^{q+1} l_i \\
&\qquad\qquad\qquad\qquad\qquad\qquad\qquad\qquad \text{(substitute } u_i \text{ with } \tilde{l}_i) \\
&= (\sum_{i=0}^{p-2} + \sum_{i=p-1}^{p+1})(1) + \sum_{i=0}^{p-2}(-\tilde{l}_i) + \sum_{i=q+2}^{K} l_i + \sum_{i=p+2}^{q-2} \theta + \sum_{i=p-1}^{p+1}(-\tilde{l}_i) + \sum_{i=q-1}^{q+1} l_i \\
&\qquad\qquad\qquad\qquad\qquad\qquad\qquad\qquad \text{(Rearrange the terms)} \\
&= (p+2) + \sum_{i=0}^{p+1}(-\tilde{l}_i) + \sum_{i=q-1}^{K} l_i + \sum_{i=p+2}^{q-2} \theta \\
&\qquad\qquad\qquad\qquad\qquad\qquad\qquad\qquad \text{(Sum all the 1's)}
\end{aligned}
$$

From Lem. B.8, $\sum_{i=0}^{p+1}(-\tilde{l}_i) + \sum_{i=p+2}^{q-2} \theta + \sum_{i=q-1}^{K} l_i \in [1, 1]$ by plugging in $-\tilde{l}_i, l_i, \theta$ to $\eta_i$. So,

$$L_1' \in [p+1, \; p+3]$$

$$
\begin{aligned}
U_1' &= \sum_{i=0}^{p-2} u_i + \sum_{i=q+2}^{K} l_i + \sum_{i=p+2}^{q-2}(1-\theta) + \sum_{i=p-1}^{p+1} u_i + \sum_{i=q-1}^{q+1} l_i & \text{(sum of right bound)} \\
&= \sum_{i=0}^{p-2}(1-\tilde{l}_i) + \sum_{i=q+2}^{K} l_i + \sum_{i=p+2}^{q-2}(1-\theta) + \sum_{i=p-1}^{p+1}(1-\tilde{l}_i) + \sum_{i=q-1}^{q+1} l_i & \text{(substitute } u_i \text{ with } \tilde{l}_i) \\
&= (q-1) + \sum_{i=0}^{p+1}(-\tilde{l}_i) + \sum_{i=q-1}^{K} l_i + \sum_{i=p+2}^{q-2}(-\theta) & \text{(sum all the 1's)}
\end{aligned}
$$

From Lem. B.8, $\sum_{i=0}^{p+2}(-\tilde{l}_i) + \sum_{i=p+2}^{q-2}(-\theta) + \sum_{i=q-1}^{K} l_i \in [-1, 1]$. Thus,

$$U_1' \in [q-2, \; q]$$

$$
\begin{aligned}
L_2' &= (p-1)(1-\theta) & \text{(sum of left bound)} \\
&= (p-1) + (p-1)(-\theta) & \text{(rearrange terms)}
\end{aligned}
$$

Because $\theta \leqslant \frac{1}{K+1}$, and $-K \leqslant p - 1 \leqslant K$, we have $(p-1)(-\theta) \in [-1, 1]$. Hence,

$$L_2' \in [p - 2, \; p]$$

$$
\begin{aligned}
U_2' &= (p-1) + (K - q - 1)\theta + (q - p - 3) + 3 + 3 && \text{(sum of right bound)} \\
&= (p - 1 + q - p - 3 + 3 + 3) + (K - q - 1)(\theta) && \text{(rearrange terms)} \\
&= q + 2 + (K - q - 1)(\theta) && \text{(sum all the 1's)}
\end{aligned}
$$

Because $\theta \leqslant \frac{1}{K+1}$, and $-K \leqslant (K - q - 1) \leqslant K$, we have $(K - q + 1)(-\theta) \in [-1, 1]$. Then,

$$U_2' \in [q + 1, \; q + 3]$$

**Step C: Analyze the bound.** It remains to show that $l - \delta \leqslant L_2 \leqslant L_1 \leqslant l + \delta$ and $u - \delta \leqslant U_1 \leqslant U_2 \leqslant u + \delta$.

Recall that we have set that $\delta = 3\tau$. Also $l \in [p\tau, (p+1)\tau)$, then

$$l - \delta < (p - 2)\tau \qquad \text{and} \qquad l + \delta \geqslant (p + 3)\tau$$

Since $u \in [q\tau, (q+1)\tau)$, then

$$u - \delta < (q - 2)\tau \qquad \text{and} \qquad u + \delta \geqslant (q + 3)\tau$$

We have just analyzed $L_1'$, $L_2'$, $U_1'$ and $U_2'$ above. Now we have:

$$
\begin{aligned}
L_1 = \tau L_1' \leqslant (p + 3)\tau && \qquad L_2 = \tau L_2' \geqslant (p - 2)\tau \\
U_1 = \tau U_1' \geqslant (q - 2)\tau && \qquad U_2 = \tau U_2' \leqslant (q + 3)\tau
\end{aligned}
$$

It follows from the above inequalities that

$$\boxed{l - \delta} \; < \; (p - 2)\tau \; \leqslant \; \boxed{L_2} \; \leqslant \; \boxed{L_1} \; \leqslant \; (p + 3)\tau \; \leqslant \; \boxed{l + \delta}$$

and

$$\boxed{u - \delta} \; < \; (q - 2)\tau \; \leqslant \; \boxed{U_1} \; \leqslant \; \boxed{U_2} \; \leqslant \; (q + 3)\tau \; \leqslant \; \boxed{u + \delta}$$

This concludes the proof for the case where $q \geqslant p + 3$.

**Excluded case.** Previously, we have shown that Terms 1, 3, and 5 are almost precise. The imprecise terms can only come from Terms 2 and 4. If $q \leqslant p + 2$, the only analyses that will be affected are those of Terms 2 and 4. Since $q \leqslant p + 2$, we have $p + 1 \geqslant q - 1$, which means Terms 2 and 4 have potentially less sub-terms in this case. Thus imprecise terms are less than the $q \geqslant p + 3$ case and we can apply the same analysis as above and derive the same bound.

We have thus shown that the neural network $N$ that we construct $\delta$-abstractly approximates $f$, and therefore the AUA theorem is true.

$\square$

