# OpenReview forum: "Generalized Universal Approximation for Certified Networks"
_ICLR.cc/2021/Conference — Reject_

### Official Review · AnonReviewer1 · 2020-10-23

**Rating:** 5
**Confidence:** 3

**Review:**

This paper studies the universal approximation of robust networks called the abstract universal approximation. While the traditional universal approximation aims to approximate the single output corresponding to each input value, abstract universal approximation studies the output interval generated by the input interval (or box) and the interval value propagation. The main contribution of the paper is to extend the result of Baader et al., 2020 to networks using general squashable activation functions.

I think that the main weakness of this paper is its novelty. The main idea of the proposed result seems very similar to that by Baader et al., 2020: approximate a target function using a summation of indicator functions. This idea and using a squashable (or sigmoid) activation function to approximate the indicator function have been widely used in the universal approximation literature. Hence, I think that the result of this paper (Theorem 4.2) can be viewed as a simple extension of the result by Baader et al. 2020.

The authors additionally claim that their construction and analysis are simpler than those by Baader et al., 2020. However, I think that this is only a minor improvement as the main focus is to show the 'existence' of networks, approximating functions under constraints.

Due to these reasons, my decision tends to reject.

---

> ### Author Response · Authors · 2020-11-12
> **Response to the lack of novelty**
>
> Thank you for your review.
>
> Please see our response about novelty in the official comment section. As we indicated above, the normal semantics and the interval bound of a function can be very different, therefore the construction and analysis of the indicator function is much more complicated. For example, constructing and analyzing the indicator function is the most technically complicated and involved part in Baader et al.[2020]. Hence we respectfully disagree that this is a simple extension.
>
> We would appreciate further feedback.

---

### Official Review · AnonReviewer2 · 2020-10-28
**Incremental Result**

**Rating:** 4
**Confidence:** 5

**Review:**

This paper proposes to extend the techniques of Baader et al. [2020], demonstrating that interval analysis provable ReLU networks are universal approximators, to a larger class of activation functions, which they call squashable functions.    Furthermore, they claim that their proof of this theorem is simpler due to using a bounded depth construction.

Pros:

* Their theorem does reduce the depth of the necessary construction.

* Expanding the theorem to more activation types is a potentially useful contribution.

* Technically, the paper appears to be correct.

Cons:

* The proposed method mostly follows the same proof technique ad Baader et al. [2020] and thus is not particularly novel.

* The authors claim that their method uses the most commonly used activation functions, yet ReLU is by far the most relevant activation function and is already included in the original theorem.

* The number of ReLUs they use for an indicator function (they might need to use intractably many indicator functions) is asymptotically identical to Baader et al [2020]. They have only reduced the number of neurons used per indicator by a constant.  However, it is unclear whether this is an advantage, since the method appears to use more slices as well.  How do the two methods compare in terms of number of neurons for the entire net when considering the same function, delta, and epsilon?

* The title “Generalized Universal Approximation for Certified Networks” is claims a significant improvement in generality over the original paper, Universal Approximation with Certified Networks (Baader et al. [2020]), yet the addition of other activation functions is hardly a significant enough difference to warrant such a claim of generality.

* Care is not taken when introducing concepts from prior work:  Section 3.2 discusses concepts introduced first by Cousot et al. [1992] and then brought to neural networks by Gehr et al. [2018], but is presented as if it were new.  Definition 4.1 is an idea developed by Baader et al [2020] yet this is also not made clear.  Section 6.2 proposes an idea that is very similar to Def 4.2 from Baader et al and uses concepts introduced as the nodal basis function in He et al [2018] but does not make this clear.

Final Review:

In summary, while technically the paper does not appear to have flaws, its contributions are incremental, and its similarity to prior work is such that it is hard to recommend acceptance.   I am thus giving this paper a rejection.

[1] P. Cousot and R. Cousot. Abstract interpretation frameworks. Journal of Logic and Computation, 2(4):511–547, 1992.
[2] Gehr, T., Mirman, M., Tsankov, P., Drachsler Cohen, D., Vechev, M., and Chaudhuri, S. Ai2: Safety and robustness certification of neural networks with abstract interpretation. In Symposium on Security and Privacy (SP), 2018.
[3] Juncai He, Lin Li, Jinchao Xu, and Chunyue Zheng. ReLU Deep Neural Networks and Linear Finite Elements. arXiv preprint arXiv:1807.03973, 2018.

---

> ### Author Response · Authors · 2020-11-13
> **response to incremental results**
>
> Thank you for your review.
>
> We have added a response to incremental results in the official comment section. The remaining issues will be addressed here:
>
> 1. Regarding the size of the entire net. In Baader et al [2020], the function is sliced with $\delta/2$ and we used $\delta/3$ because the we need extra tolerance for error introduced by the limit of the squashable activation functions. If the error was $0$, then $\delta/2$ would be enough and the final inequality is sharp. However, because of the limit of the activations, we only need arbitrarily positive extra tolerance. For example, we can slice the function using size $\delta/(2.00000001)$ and still be able to prove the inequality by appropriately selecting other parameters. If we only consider ReLU, we can use $\delta/2$ because the extra tolerance needed in this case is $0$. As a result, the size of the entire net is a minor concern. It would be awkward to present the slice in this way in the paper so we used $\delta/3$, which appears more natural and general.
>
> 2. We used the same framework as Baader et al [2020] and we have pointed out several times in the paper, including using the indicator. The idea of using the nodal basis function was already mentioned in Baader et al [2020], and it is a very common idea in the finite element method. The ideas and concepts are only high-level similar, and would not imply our construction and proof.  Def 4.2 from Baader et al. appears similar to (3) structually (i.e. using an activation to evaluate over the information from all dimensions), but none of their constituents are the same (i.e. different activations and in def 4.2 inside the activation it is the 2m-ary min function, while in (3) it is the sum over all 1-dimensional indicator functions.). We will clarify all standard concepts due to Cousot and Cousot and their adaptation by Gehr for neural networks, and cite works that introduced similar ideas in the updated paper.
>
> Thanks again for the detailed review and we would appreciate further feedback.

---

### Official Review · AnonReviewer4 · 2020-10-28
**Universally approximating continuous functions by neural networks**

**Rating:** 5
**Confidence:** 2

**Review:**

This work studies the task of universally approximating continuous functions by (certain classes of) neural networks. The paper shows that for a continuous function $f$ there exists a neural network $N$ such that for any box $B$ the range of outputs of $N$ is close (in a parameter $\delta$) to the range of outputs of $f(x)$, $\forall x \in B$.
This proof is constructive and applies to $N$ using ReLU, sigmoid, tanh or ELU activation functions. (The result actually includes an even larger class of activation functions, that the authors call _squashable_.)

A previous work, Baader et al., shows the same result when neural networks use ReLU activation functions. The authors of this paper provide a result applicable to a broader class of networks. They also improve the size of neural networks constructed in Baader et al. (as discussed in the last paragraph of Section 6), but at many points in the paper this result feels to be incremental.

-- Minor comments about the paper:

Figure 1, which assumes Eq (1), could be moved to the top of Page 4, after Eq (1) is stated.

Sections 3 and 4 have lots of text in bold. Many paragraphs are named, and then they have definitions with the same name. It reduces readability.

The bottom of Figure 6 and the text are too close to each other.

**Updates**:

After carefully reading comments of the other reviewers as well as the authors' response, I change my score from 6 to 5.

---

> ### Author Response · Authors · 2020-11-12
> **Updated paper**
>
> Thank you for your review.
>
> We have updated our paper accordingly and added a response to the issue of incremental results in the official comment section. We would appreciate feedback.

---

### Official Review · AnonReviewer3 · 2020-10-29
**Interesting Theoretical Result on Expressivity/Certifiability of NN, but only Incremental Extension - Lack of Novelty**

**Rating:** 4
**Confidence:** 4

**Review:**

The paper shows an "augmented" universal approximation (UA) result for neural networks that the authors call Abstract UA (AUA for short) and the motivation comes from understanding expressivity and certifiability of NN. Their result holds for NN with a wide variety of activation units and this is the main point of the paper, which directly extends the same result for ReLU networks (Baader et al. 2020).

Typically, results in UA state that any continuous function f on a bounded domain can be expressed as a combination of activation units, sometimes even with only one hidden layer and one output layer, which is the classical result by Cybenko (1989), Hornik (1989) and more.  Their augmented version asks what if we want to have a more robust version of those statements, where we want to certify that any given box in the domain of f is mapped through the NN in such a way so that the NN values always stay close to the values of f. More formally, for some given accuracy \delta, we want to have |NN(x) - f(x)| < \delta for all x in the box. Furthermore, we would like to be able to certify this property and to do so we can use the technique of interval propagation. (To put it differently, the image of any interval of f is basically "sandwiched" between two close-by surfaces that are output by the NN.)

The result is of theoretical nature and this is fine as expressivity and certifiability are not yet well-understood. The major issue with the paper is the lack of novelty. The most related paper in terms of techniques and ideas (Baader et al.) proved the exact same result when the activations are ReLUs instead of the wider variety of activation units considered in the paper. The units considered here, are basically monotone functions that have upper and lower cut-offs or can be phrased as such after a simple transformation. The main ideas for approximating functions via step functions and indicators on intervals or boxes has been used for decades (even starting with the folklore results in approximation theory) and there seems to be lack of substantially new ideas. The reviewer feels that given the previous work on ReLUs (Baader et al.), the proof for the more general activation units can be reverse-engineered. Finally, of secondary importance the authors state some simplifications over Baader et al.

A minor issue that the reviewer is ok with, is that for a more practice-oriented person, it is hard to buy the motivation from adversarial robustness/certifiability as in many settings the adversary will not just choose an attack bounded in \ell_infinity by some small parameter. It is nice to have certifiable NN and know their behaviour if the inputs slightly change, however the motivation also needs to address what happens when a few large perturbations are allowed like the single-pixel attack ("One pixel attack for fooling deep neural networks").

Overall, conditioned on the immediate prior work,  the result itself simply extends AUA from ReLU activations to more general units, and is not surprising.

One quick suggestion: Can the authors clarify how the lipschitzness of f or of the NN can affect the statements? Otherwise, the theorem may be misinterpreted as a consequence of continuity.

Suggestions for future improving:
Did the authors consider proving something along the lines of depth separation results for such kinds of AUA? The reviewer believes this could give a new dimension to the AUA theorems and draw another connection with certifiability and depth/width tradeoffs. Depth is usually preferred in practice over width, and a formal way for proving this is to show the power of depth in representing functions: for example, relevant works here are Telgarsky's (Benefits of depth in neural networks) and later improvements (Better Depth-Width Trade-offs for Neural Networks through the lens of Dynamical Systems) that show exponential separations in depth vs width by constructing highly oscillatory functions. However these works do not consider certifiability which may be an opportunity for your techniques.

Another quick question: Is there a formal comparison of your squashable activation units to the semi-algebraic units considered in Telgarksy? Maybe your results can also capture such activations?

Other comments:

Theorem 3.3:  Do the authors consider this one of the main contributions of the paper? Wouldn't it be better to be stated as a proposition, as it is quite straightforward for the activation units stated there?

p1: "Their theorem states that not only can neural networks approximate any continuous function f
(universal approximation)" ... the reviewer believe that this first part of the sentence is a bit misleading as universality has been known since the 80s. The sentence starting with "but..." is the extension presented in Baader et al. and that is the contribution, right?

Fig.1: When one takes interval bounds and considers a superset of box B, shouldn't the output be superset of f(B)? Maybe the figure is a bit confusing? What does it mean:  "(Right is adapted from Baader et al. (2020).)"

p2: When Abstract universal approximation (AUA) theorem is stated: Again f is continuous so it should be clarified. Also: Line 5 of this paragraph: interval bounds are stated but if the parameters are not given (is it \eps?) like it was done in the intro, then the result is not meaningful as it would be a consequence of the continuity of f and Lipschitzness of NN (please discuss).  Maybe another way to phrase the same thing, is to say that whenever the input is changed to B+\eps, the network follows f(B)+\delta for appropriate parameters? The point is just that the NN will not do anything "too wild" correct? In other words, there is a NN that will map intervals of f to not-much-wider intervals correct?

Typos/Inaccuracies:
p2: the AUA theorem implies us that
p2: while controlling approximation error δ
p2: a bounded number of layers --> just one hidden layer was proved in Cybenko. What is the "standard" feedforward NN?
p2: an universal -> a (several places)
p6: has rigid values --> rigid? (meaning relu is fixed everywhere?)

---

> ### Author Response · Authors · 2020-11-16
> **Response**
>
> Thanks for your insightful review.
>
> Please see our response about novelty in the official comment section. We believe that the result is a necessary extension because otherwise the AUA theorem seems limiting that we only have it for the ReLU network, contrary to the UA theorem. Technically speaking, even though using summation of indicator function is a common idea, because the interval bound propagation is different from the normal semantics of the function, its construction and analysis could be quite challenging compared to the normal approximation theory or mathematical analysis. The construction and analysis of the indicator function is in fact the most technical part in Baader et al. [2020]. Below we will respond to other comments:
>
> How about one-pixel attack?
> From a purely theoretical point of view, in finite dimensional space all the p-norm induced metric spaces are strongly equivalent. Using any p-norms ($p\geq 1$) would not influence the theorem statement itself. However, a few large perturbations will violate the $\epsilon$-ball requirement under the $L_\infty$ space. That is to say, even though the network is $\epsilon$-certifiably robust under the $L_\infty$ space, it does not imply that it is robust to the one-pixel attack. In practice, one can still use interval propagation to certify whether a network is robust to the one-pixel attack. That would be a trade-off between accuracy and efficiency. To achieve an accurate robustness certificate, we might have to practice the interval bound on the pixel one-by-one. One can still apply the interval bound over all the pixels at once, but we might not be able to obtain a robustness certificate.
>
> How the Lipschitzness of f or of the NN can affect the statements?
> The Lipschitzness of f in the AUA theorem is an orthogonal issue. That is to say, qualitatively we can approximate any continuous function in a compact domain. However, if the function has a small Lipschitz constant, we will need less indicator functions to construct the approximation as in the usual approximation theory cases because the local variation of the function is small.
>
> Comparison with semi-algebraic units: Semi-algebraic and squashable activations are two different sets of functions. Semi-algebraic functions are essentially finite algebraic operations over polynomials, which apparently cannot capture the canonical sigmoid function; similarly polynomial functions are semi-algebraic but not squashable. This is due to that with bounded depth and arbitrary width, neural networks with polynomial activations effectively computes a polynomial up to certain finite degree, and it cannot approximate any functions in the pointwise sense [Leshno et al. 1993]. Because AUA is stronger than UA, AUA cannot hold with polynomial activations. Again the interval propagation can be more difficult than the normal semantics analysis of a function. As we pointed out in the general comment, constructions that work in the normal semantics might not work under the interval bound. Another more practical issue is that arbitrarily abstract interpreting a function is very hard. For example, if we would use semi-algebraic function as the activation function, we might not be able to work with even a single such function, because we should know precisely what are the maximum and minimum values of the function on an arbitrary interval but most of the activation functions are non-convex, and it is impractical to know these values. For Telgarsky's work, which heavily used oscillatory functions, its interval bound propagation is infeasible to analyze.
>
> Other comments:
> We have updated the paper accordingly. Thanks for pointing many things out. We used theorem 3.3 because it is a non-trivial construction when we need to take its interval bounds into account as stated in the general comment.
>
> For fig. 1, the input of the interval bounds of N is B, not a superset of B, and its output is an interval in R. Baader et al. used a similar plot in their paper to illustrate the theorem.
>
> The theorem statement is not related to $\epsilon$. We use $\epsilon$ when we construct the approximation. I think there is a small misunderstanding on the interval bound propagation. Again even if a NN is continuous or Lipschitz, its interval bound is not necessarily the same as its normal semantics (see our example in the general comments section). The intuition regarding the statement is correct, but we have to construct the NN that admits the interval bound. Essentially the interval bound propagation is a calculus which will automatically output an interval, rather than magically guess the upper and lower bounds of a function.
>
> When we use standard feed-forward NN we mean the neural network constructed using the grammar defined in 3.1. We were sloppy when using rigid as in many mathematical usages of rigidity. Essentially it means ReLU is a fixed function.
>
> Thanks again for the detailed comments and we appreciate further feedback.

---

### Author Response · Authors · 2020-11-12
**Response to limited novelty**

A number of reviewers pointed out the limited novelty. Below, we argue that both our result and proof construction are novel and non-trivial, requiring a sharp departure from existing work.

At the result level:

1. This work is a strong generalization of Baader et al. [2020], the AUA (abstract universal approximation) theorem built exclusively with ReLU activations. It is a natural question to ask how broad the AUA theorem can be applied, in a spirit similar to how generations of researchers sought to understand the limits of the UA theorem. It seems limiting that the AUA theorem only works for ReLU networks.

2. A key novelty comes from the fact that we discovered a broad class of functions, called squashable functions, which includes most practically used activation functions, for which AUA holds. It was not clear that some of the functions were indeed squashable under the interval bounds. For example, if we consider ReLU, one might propose that t’(x) = ReLU(x+1)-ReLU(x) satisfies equation (1) (page 3). However, this is only true under the normal (set) semantics. Under the interval bounds,  t’([10,20]) = ReLU([11, 21]) - ReLU([10,20])=[11, 21] - [10,20]=[-9,11]. So t’ is not a squashable function under the interval bounds. We have provided a construction (see Proposition 3.3) that transforms those ReLU-like functions, and their interval bounds indeed satisfy equation (1) as we proved.

3. We demonstrate the existence of certifiably $L_\infty$ robust neural networks as a result of the AUA theorem (section 4). Together with the work on the activation functions, our result lays the theoretical foundation on why interval-bound propagation can certify the robustness of neural networks with various activation functions.

For the constructive proof, we use the summation of indicator functions as the framework to build the approximation. Even though this is a classical idea in approximation theory or even mathematical analysis at large, it is far from simple to build and analyze the interval bound of the indicator function.

1. In Baader et al. [2020], constructing and analyzing the indicator function is the most technically challenging part, i.e., constructing the nmin function using ReLU and then the 2m-ary nmin function. Its analysis is also very sophisticated because even when the normal semantics of the 2m-ary indicator function are clear it is still hard to analyze its interval bound. Therefore, a majority of the technical content in Baader et al. [2020] is devoted to the construction and analysis of the indicator function. (See pages 5 and 6, and from definition A.4 to Lemma A.17 in the appendix of https://openreview.net/pdf?id=B1gX8kBtPr)

2. Because we are considering a much more general set of activations, our construction is necessarily distinct from Baader et al. [2020]. We start from observing that the squashable function can approximate the sign function and then a one-dimensional indicator function (section 6.1). We simulate the logical OR operation by summing m one-dimensional indicator function to construct the m-dimensional indicator function (section 6.2).

3. One advantage of our construction is that its interval bounds closely resemble normal semantics. If we know the properties of the function, we can immediately understand its interval bounds (see sections B.3 and B.4 in the appendix). Thus, not only the construction itself is simpler, but also its interval-bound analysis.

---

### Decision · Program_Chairs · 2021-01-07
**Final Decision**

**Decision:**

Reject

**Comment:**

The authors extend prior work showing that networks trained to be certifiably robust using interval bound propagation are universal approximators. They extend prior results applicable to ReLU networks to a much broader class of networks with general activation functions.

The paper makes an interesting contribution to the literature relative to prior literature showing that one need not sacrifice universal approximation guarantees while training networks with IBP to be certifiably robust to l_inf attacks.

Since the paper is primarily theoretical, the main concern raised by the reviewers was around novelty and the theoretical significance of ideas presented relative to prior work. While proof techniques may be novel, the extension of AUA results to alternate activation functions is not surprising and do not substantially contribute to the field's understanding of learning certifiably robust networks particular since most SOTA results for IBP-based training have been achieved with ReLU based networks. The authors' rebuttal did not providing convincing arguments for the reviewers to revise their scores. Hence I do not feel that the contributions of the paper justify acceptance at this time.